

# Identifying cloud droplets beyond lidar attenuation from vertically-pointing cloud radar observations using artificial neural networks

Willi Schimmel[1], Heike Kalesse-Los[1], Maximilian Maahn[1], Teresa Vogl[1], Andreas Foth[1], Pablo Saavedra Garfias[1], and Patric Seifert[2]

[1]Institute for Meteorology (LIM), Leipzig University, Leipzig, Germany
[2]Leibniz Institute for Tropospheric Research (TROPOS), Leipzig, Germany

**Correspondence:** Willi Schimmel (willi.schimmel@uni-leipzig.de)

**Abstract.** In mixed-phase clouds, the variable mass ratio between liquid water and ice as well as the spatial distribution within the cloud plays an important role for cloud life time, precipitation processes, and the radiation budget. Data sets of vertically-pointing Doppler cloud radars and lidars provide insights into cloud properties at high temporal and spatial resolution. Cloud radars are able to penetrate multiple liquid layers and can potentially be used to expand the identification of cloud phase to the

entire vertical column beyond the lidar signal attenuation height, by exploiting morphological features in cloud radar Doppler spectra that relate to the existence of supercooled liquid. We present VOODOO (reVealing supercOOled liquiD beyOnd lidar attenuatiOn), a deep convolutional neural network (CNN)-based retrieval mapping radar Doppler spectra to the probability for the presence of cloud droplets (CD). The training of the CNN was realized using the Cloudnet processing suite as supervisor. Once trained, VOODOO yields the probability for CD directly at Cloudnet grid resolution. Long-term predictions of 18 months

in total from two mid-latitudinal locations, i.e. Punta Arenas, Chile (53.1°S, 70.9°W) in the Southern Hemisphere and Leipzig, Germany (51.3°N, 12.4°E) in the Northern Hemisphere are evaluated. Temporal and spatial agreement in cloud-droplet bearing pixel is found for the Cloudnet classification to the VOODOO prediction. Two suitable case-studies were selected, where strati-form, multi-layer and deep mixed-phase clouds were observed. Performance analysis of VOODOO via classification-evaluating metrics reveals precision $> 0.7$, recall $\approx 0.7$ and accuracy $\approx 0.8$. Additionally, independent measurements of liquid water path

(LWP) retrieved by a collocated microwave radiometer (MWR) is correlated to the adiabatic LWP, which is estimated using the temporal and spatial locations of cloud droplets from VOODOO and Cloudnet in connection with a cloud parcel model. This comparison resulted in stronger correlation for VOODOO ($\approx 0.45$) compared to Cloudnet ($\approx 0.22$) indicates the availability of VOODOO to identify CD beyond lidar attenuation. Furthermore, the long-term statistics for 18 months of observations are presented, analyzing the performance as function of MWR-LWP and confirming VOODOO's ability to identify cloud droplets

reliably for clouds with LWP $> 100\,\mathrm{g\,m^{-2}}$. The influence of turbulence on the predictive performance of VOODOO was also analyzed and found to be minor. A synergy of the novel approach VOODOO and Cloudnet would complement each other perfectly and is planned to be incorporated into the Cloudnet algorithm chain in the near future.



## 1 Introduction

In mixed-phase clouds, the variable mass ratio between liquid water and ice as well as its spatial distribution within the cloud
plays an important role for cloud life time (Morrison et al., 2012), precipitation processes (Mülmenstädt et al., 2015), climate
feedbacks (Choi et al., 2014; Bjordal et al., 2020), and the radiation budget (Sun and Shine, 1994; Shupe et al., 2004; Turner,
2005). Modeling mixed-phase clouds is challenging and requires the basic assumption of thermodynamic phase, i.e. ice, liquid
or mixed (Zhao et al., 2012). Improving methods for cloud phase classification is a first step towards minimizing the error of
liquid water content (LWC) and particle size retrievals (Riihimaki et al., 2016). Accurately observing the phase distribution
within mixed-phase clouds has historically been one of the major challenges for the remote sensing community (Shupe et al.,
2008). Multisensor retrievals rely mostly on valid lidar signals (Shupe et al., 2005; Illingworth et al., 2007; de Boer et al., 2009;
Silber et al., 2020) in synergy with Doppler cloud radar moments, limiting the liquid classification due to lidar attenuation
(Shupe et al., 2004; Sokol et al., 2018). The lidar signal is strongly attenuated by liquid water layers with optical depths $\tau \sim 3$
$- 5$ (Silber et al., 2020), hampering the use of lidar-based hydrometeor target classification for optically thick clouds. Radars
are able to penetrate multiple liquid layers and can thus be used to expand the identification of cloud phase to the entire vertical
column beyond the lidar signal attenuation height, if morphological features in cloud radar Doppler spectra can be related to
the existence of supercooled liquid droplets.

Several efforts have been made in the past to exploit these features and derive the distribution of liquid in mixed-phase
clouds: Continuous-Wavelet transformations in combination with fuzzy logic using fixed thresholds to identify liquid peaks in
simulated Doppler spectra have e.g. been employed by Yu et al. (2014). Doppler spectra peak finding algorithms like PEAKO
(Kalesse et al. (2019); a supervised learning method) and peakTree (Radenz et al. (2019); a binary tree approach) can be used
to identify the number of radar Doppler spectra peaks at each time-height step. In the next step, individual peaks can be related
to liquid droplet existence based on the sub-peak radar moment analysis as done in Radenz et al. (2019). Silber et al. (2020)
applied statistical tests to cumulative distribution functions and probability density functions of radar moments and temperature
measurements by soundings to discriminate liquid-bearing from pure-ice cloud layers. Kalogeras et al. (2021) presented another
method that used climatologically derived, per-phase probability distributions to retrieve a ice/liquid partitioning via a per-
pixel, neighborhood-dependent algorithm. Also, deep learning approaches have been used in the past to derive a liquid mask
from lidar backscatter coefficient and depolarization predicted from Doppler radar spectra (Luke et al., 2010). Although the
applicability of machine learning to cloud radar data has been demonstrated by Luke et al. (2008, 2010) and Kalesse et al.
(2019), its potential is far from being fully exploited. The aim is to develop a robust method which is able to directly relate
raw Doppler spectra information to the presence of liquid hydrometeors, without the need for complex feature engineering and
extraction.

The interest in machine learning and particularly deep learning in the Earth system sciences has strongly increased in the
past few years (Maskey et al., 2020). Deep learning techniques are a subset of machine learning, where deep artificial neural
networks (ANN) learn relationships from data. These applications are particularly powerful due to their ability to perform part
of the data pre-processing themselves. Vogl et al. (2022) showed that ANNs can be used to predict riming using ground-based





zenith-pointing cloud radar variables radar reflectivity, spectrum width and skewness. An earlier approach from Luke et al. (2010) transfers the features of Doppler spectra into particle backscatter and volume depolarization of a high spectral resolution lidar (HSRL) using a multi-layer perceptron model and was further validated by Kalesse-Los et al. (2022) by applying the pre-
trained machine learning model to data from the Analysis of the Composition of Clouds with Extended Polarization Techniques (ACCEPT) campaign (Myagkov et al., 2016a, b). The methods mentioned above rely mostly on fixed thresholds for radars (i.e. reflectivity, mean Doppler velocity, spectrum width, skewness, spectrum edge slopes) and lidars (i.e. attenuated backscatter and depolarization) and are only applicable for a small subset of cloud types (Luke et al., 2008, 2010; Yu et al., 2014; Kalesse et al., 2019; Silber et al., 2020). Nevertheless, in the study of Kalesse-Los et al. (2022), the Luke et al. (2010) model displayed
the ability to perform well on Doppler spectra recorded by a different cloud radar in different atmospheric conditions.

In this study we build upon the idea of Luke et al. (2010) and use a deep convolutional neural network (CNN) model to predict directly a probability of the distribution of supercooled cloud droplets in mixed-phase clouds observed by a vertically-pointing Doppler cloud radar. Relevant spectral signatures such as bi-modalities, spectral skewness, and temporal evolution can be extracted by a deep CNN that relates to the cloud phase by training in a supervised scheme, using Cloudnet's target
classification as supervisor. As part of the pan-European Aerosol, Clouds and Trace Gases Research Infrastructure (ACTRIS), the Cloudnet processing suite (Illingworth et al., 2007; Tukiainen et al., 2020) is tailored to process observations and model data on the composition of the atmosphere. The measurements and model data are brought on a common grid and the targets are classified as ice, liquid, aerosol, insects, among others. Here, the information about the presence of liquid droplets is extracted from the Cloudnet target classification and in a first step used to train VOODOO. In the next step, Cloudnet data which were
not used for training are compared to the predictions of VOODOO. Various binary classification metrics are used to quantify performance, such as precision, recall, accuracy and F1-score. Further evaluation is done by correlating several independent measurements such as liquid water path (LWP) retrieved by microwave radiometer as suggested by Luke et al. (2010) and Kalesse-Los et al. (2022).

The paper is structured as follows. Section 2 gives an overview about the instrumentation and the data sets used in the context
of this work. In Section 3 the methodology of the VOODOO retrieval is presented. Section 4 is divided into the analysis of a case study and the statistical evaluation of VOODOO by application to a total of 18 months of measurement data from two different geographical locations. The paper concludes with a summary and outlook in Section 5.

## 2   Instrumentation and data set

This section introduces the instrumentation used to train and validate the CNN performance. First, the data sources are pre-
sented (Sec. 2.1), followed by a short description of the two field experiments in Punta Arenas and Leipzig, including specifics of the respective sites (Sec. 2.2).



**Table 1.** Specifications of instruments/models and measured/modeled quantities used in this study.

| Data source (Reference) | Frequency $\nu$ Wavelength $\lambda$ | Measured / retrieved quantity | Temporal resolution | Vertical range | Vertical resolution |
|---|---|---|---|---|---|
| **Doppler cloud radar** RPG-FMCW-94-DP (Küchler et al., 2017) | $\nu = 94\,\mathrm{GHz}$ | Spectral power $S(v_D)$ Radar reflectivity factor $Z_e$ Mean Doppler velocity $\bar{v}_D$ Spectrum width $\sigma_w$ Linear depolarization ratio LDR | 5 s | $120 - 12000\,\mathrm{m}$ | $30 - 45\,\mathrm{m}$ |
| **Microwave radiometer** RPG-HATPRO-G2 (Punta Arenas) RPG-HATPRO-G5 (Leipzig) (Rose et al., 2005) | $\nu = 22.24 - 31.4\,\mathrm{GHz}$ $\nu = 51.0 - 58.0\,\mathrm{GHz}$ | Brightness temperatures Liquid water path LWP | 1 s | column integral | |
| **Ceilometer** Jenoptik CHM15kx (Punta Arenas) Lufft CHM15k Nimbus (Leipzig) (Heese et al., 2010) | $\lambda = 1064\,\mathrm{nm}$ | Attenuated backscatter coefficient $\beta_{\mathrm{att}}$ | 30 s | $15 - 15000\,\mathrm{m}$ | 15 m |
| **Weather model forecast** ECMWF (Owens and Hewson, 2018) | | Temperature $T$ Pressure $P$ Relative Humidity HUM | 3600 s | $10 - 12000\,\mathrm{m}$ | $20 - 300\,\mathrm{m}$ |

## 2.1 Data source

Four data sources (see Table 1) are considered for the presented CNN retrieval: cloud radar Doppler spectra from a vertically-pointing radar, attenuated backscatter coefficient $\beta_{\mathrm{att}}$ from a ceilometer, liquid water path (LWP) retrieved from a microwave

radiometer (MWR), and temperature, relative humidity and pressure from numerical weather forecast data from the European Centre for Medium-Range Weather Forecasts (ECMWF).

Doppler cloud radars record the distribution of reflectivity in the Doppler velocity domain $(v_{\mathrm{min}}, v_{\mathrm{max}})$ continuously in time and provide vertically resolved observations with excellent sensitivity to small hydrometeors (Kollias et al., 2007). Fig. 1 shows hydrometeors creating peaks at distinct terminal fall velocities in the Doppler radar spectra, the larger and heavier they are, the

faster they fall (assuming no vertical air motion). The radar reflectivity factor $Z_e$ in the Rayleigh scattering regime is sensitive to the sixth power of the diameter, $Z_e \sim N \cdot D^6$, where $N$ is the the number of hydrometeors for diameter $D$. It follows, $Z_e$ is larger for small populations of large hydrometeors such as ice crystals or drizzle and rain, thus dominating the spectral signal by producing large peaks at higher fall velocities. In contrast, cloud droplets exist in the atmosphere with much larger $N$ but much smaller $D$ at $\approx 0\,\mathrm{m\,s^{-1}}$ terminal velocity, producing low intensity peaks in the Doppler spectra. The radar moments, i.e.

reflectivity factor $Z_e$, mean Doppler velocity $\bar{v}_D$, and spectral width $\sigma_w$, and linear depolarization ratio LDR are computed from the Doppler spectra. These radar moments are required for Cloudnet processing and are usually sufficient to derive a mask for precipitation and ice crystals (Illingworth et al., 2007). However, these radar moments alone are not in all situations sufficient to provide the necessary information content to characterize liquid and mixed-phase clouds reliably.

The ceilometer provides high resolution profiles of attenuated backscatter coefficient $\beta_{\mathrm{att}}$ at 1064 nm. The parameter $\beta_{\mathrm{att}}$

is sensitive to the second power of the diameter, $\beta_{\mathrm{att}} \sim N \cdot D^2$. It follows that, $\beta_{\mathrm{att}}$, is larger for large populations of small



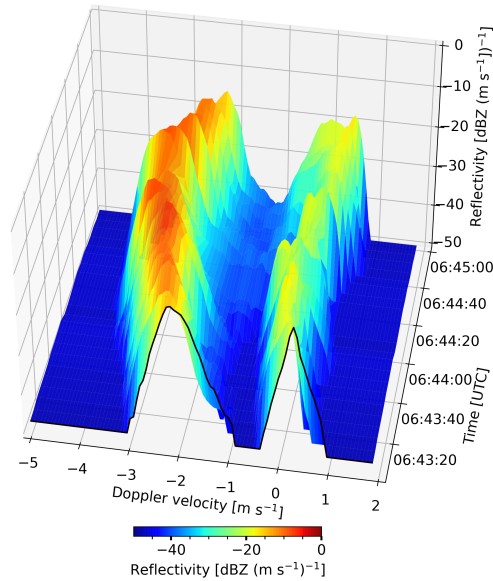

**Figure 1.** Time-spectrogram from Aug. 1, 2019 at 2400 m altitude in Punta Arenas, Chile. The bi-modal Doppler-spectra contain liquid cloud droplets the right-hand-side of the spectrum (slow moving) and ice crystals on the left side of the spectrum (faster moving). Cloudnet classification result for this particular time-range slice is "ice and supercooled liquid droplets" at -10 °C.

hydrometeors (cloud droplets). Small amounts of larger ice crystals, drizzle and rain drops return a much lower signal. For this reason, $\beta_{\mathrm{att}}$ is used to distinguish between cloud droplets and aerosol in Cloudnet.

The MWR measures brightness temperature profiles over a band of different frequencies. To derive the LWP and integrated water vapor (IWV), Optimal Estimation (Foth and Pospichal, 2017) or artificial neural network retrievals (Yan et al. (2020) or MWR manufacturer) are used. The LWP is a measure for the amount of liquid water in the column above the instrument and used for validation purposes in the frame of this work. The MWR-LWP is correlated to the LWP derived from VOODOO predictions using an adiabatic cloud parcel model by Karstens et al. (1994), as illustrated in Kalesse-Los et al. (2022).

Temperature, relative humidity and pressure profiles, taken from the European Centre for Medium-Range Weather Forecasts (ECMWF) model are used in combination with MWR-LWP to derive atmospheric profiles of gaseous and liquid attenuation for correction of the radar returns used within the Cloudnet processing chain. The numerical weather forecast data can be downloaded via the Cloudnet data portal (https://cloudnet.fmi.fi).

## 2.2 Data sets from Punta Arenas and Leipzig

This work is based on remote-sensing measurements from two different geographical locations, i.e. Punta Arenas, Chile and Leipzig, Germany. Selected key properties of the two sites are summarized in Table 2. First, observations from the long-term





**Table 2.** Overview on location, data availability, climate, aerosol load, and related studies for the data sets used. The altitudes are given above mean sea level (asl).

| Location | Punta Arenas, Chile | Leipzig, Germany |
|---|---|---|
| | 53.1°S, 70.9°W | 51.3°N, 12.4°E |
| Station altitude | 9 m asl | 125 m asl |
| Campaign name | DACAPO-PESO | LIM |
| Measurement period | 301 d | 488 d |
| Cloudnet availability | 262 d | 400 d |
| Climate | Southern mid-latitudes | Northern mid-latitudes |
| Typical aerosol load | Marine, occasionally continental | Continental background, occasionally dust |
| Related studies | Kanitz et al. (2011) | Ansmann et al. (2005) |
| | Ohneiser et al. (2020) | Seifert et al. (2010) |
| | Bromwich et al. (2020) | Bühl et al. (2013) |
| | Jimenez et al. (2020) | Bühl et al. (2016) |
| | Floutsi et al. (2021) | Radenz et al. (2021) |
| | Radenz et al. (2021) | Vogl et al. (2022) |
| | Vogl et al. (2022) | |

field experiment Dynamics, Aerosol, Cloud, and Precipitation Observations in the Pristine Environment of the Southern Ocean (DACAPO-PESO) in Punta Arenas, Chile are discussed and second, observations from the roof platform of the main building of the Leipzig Institute of Meteorology in Leipzig (LIM), Germany are analyzed.

DACAPO-PESO focuses on the investigation of aerosol-cloud-dynamics and interactions in the atmosphere. A unique data set has been gathered by synergistic retrievals with active and passive remote sensors. Clean pristine marine air masses dom-

inate the aerosol conditions, due to almost constant westerly winds (Schneider et al., 2003; Foth et al., 2019; Jimenez et al., 2020; Floutsi et al., 2021; Radenz et al., 2021). Additionally, gravity waves have been observed frequently over Punta Arenas (Alexander et al., 2017; Silber et al., 2020; Radenz et al., 2021). Due to orographic effects induced by strong westerly winds moving over the Andes mountain range, gravity waves are a general feature in the vicinity of all landmasses in the middle and high latitudes of the Southern Hemisphere (Sato et al., 2012; Alexander et al., 2016). The Leipzig Aerosol and Cloud Remote

Observations System (LACROS) suite has been deployed by the Leibniz Institute for Tropospheric Research (TROPOS) from Nov. 27, 2018 to Nov 20, 2021. LIM contributed a RPG-FMCW94 Doppler cloud radar, operating from Nov. 27, 2018 until Sep. 27, 2019, to enhance the information content of the DACAPO-PESO field campaign.

The second data set includes measurements recorded at the roof platform of the main building of the Leipzig Institute of Meteorology (LIM). The observations included in this work were conducted from December 17, 2020 to March 6, 2022.

Leipzig is located in central Europe and is predominantly influenced by continental air masses, anthropogenic pollution (Baars





et al., 2016) and occasional mineral dust events (Seifert et al., 2010). A more in-depth analysis of aerosol contributions for both sites can be found in Radenz et al. (2021).

## 3 Methodology

This section introduces the machine learning methodology used to derive a spatio-temporal liquid cloud droplet probability
distribution directly from spectra of the vertically-pointing Doppler cloud radar. A CNN is trained on radar Doppler spectra using the atmospheric target classification retrieved by the Cloudnet algorithm (Illingworth et al., 2007; Tukiainen et al., 2020) as supervisor. An example is given in Fig. 2. Firstly, the pre-processing of Doppler radar spectra and sampling of features is presented in Sec. 3.1. Secondly, Cloudnet processing is done to derive the hydrometeor target classification reference labels for the training data, explained in Sec. 3.2. Section 3.3 describes the machine learning model, followed by introducing the
training and validation data sets in Sec. 3.4. Information about the training process is given in Sec. 3.5. Post-processing steps are described in Sec. 3.6 and validation metrics are presented in Sec. 3.7.

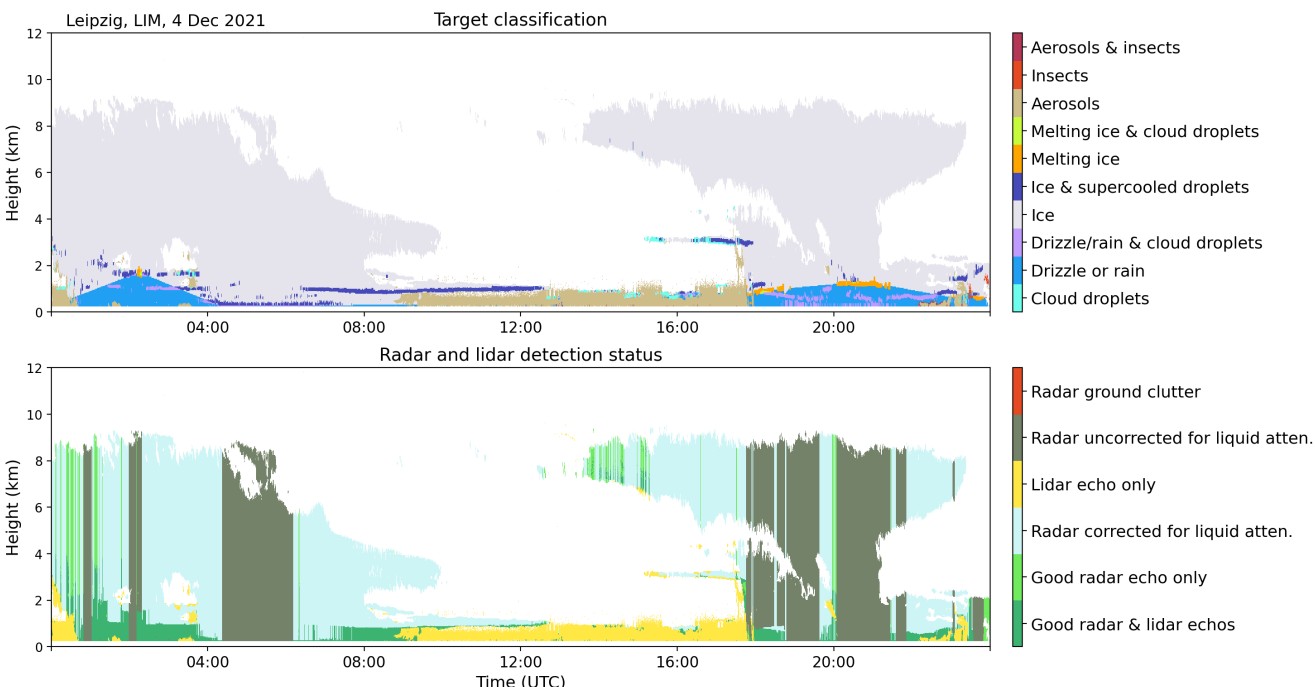

**Figure 2.** Cloudnet target classification (top) and radar-lidar detection status (bottom) for 4. December 2021 in Leipzig, Germany.





**Table 3.** Specifications and program settings for the vertically-pointing RPG-FMCW-94-DP Doppler cloud radar.

| Attributes | Chirp sequence (CS) | | |
| --- | --- | --- | --- |
| | CS 1 | CS 2 | CS 3 |
| Integration time [s] | 0.52 | 1.77 | 2.71 |
| Range interval [m] | 100 – 1200 | 1200 – 7000 | 7000 – 12000 |
| Range vertical resolution [m] | 29.8 | 44.7 | 39.7 |
| Nyquist velocity [m s$^{-1}$] | 9.0 | 6.3 | 4.7 |
| Doppler velocity bins | 256 | 256 | 128 |
| Doppler velocity resolution [m s$^{-1}$] | 0.07 | 0.05 | 0.07 |
| Averages per spectrum | 23 | 54 | 124 |
| Frequency modulation [kHz] | 300 – 3600 | 600 – 3500 | 1969 – 3375 |

## 3.1 Pre-processing and feature-sampling

Firstly, the raw radar Doppler spectra are pre-processed as follows. The RPG-FMCW-94-DP radar operates with different settings, through a user-defined measurement definition file. Table 3 contains the instrument settings which were identical for
both sites. By modulating the center frequency (94 GHz) in ranges of 300 – 3600 MHz, the atmosphere is sampled consecutively by three programs (chirps) collecting Doppler radar spectra for the three ranges of operation or chirp sequences (CS) consecutively. Each CS has slightly different range and Doppler velocity resolution, as well as different Nyquist velocity $v_{\mathrm{Nyq}}$, number of Doppler spectra bins, frequency modulation ranges and number averages (coherently and non-coherently) for a single spectrum (see Table 3).

The pre-processing steps listed below are split into general pre-processing necessary to obtain the Cloudnet products and spectral feature extraction:

- Received vertical and horizontal popularized signals are summed, yielding the total back-scattered signal intensity.

- Noise in the Doppler radar spectra is estimated and removed via manufacturer software, which uses the method of Hildebrand and Sekhon (1974). The cut-off threshold for noise removal is set at mean noise power plus six standard
deviations of the noise power.

- The radar moments $Z_e$, $\bar{v}_D$, $\sigma_w$, and LDR are estimated from the spectra. Daily files of containing these variables are stored int NetCDF files an are used as input for the Cloudnet processing.

- The Cloudnet *target classification* is derived (see Sec. 3.2).

Spectral feature extraction is done by adjusting the radar Doppler spectra for VOODOO processing.

- The presented method is designed for Doppler radar spectra counting 256 Doppler velocity bins. If the number of Doppler velocity bins does not match 256, nearest neighbor interpolation is applied to meet the number of 256 Doppler bins.





- Doppler bins which were removed from the Doppler spectra by noise filtering are replaced by their range-dependent sensitivity limit, also provided by the RPG software.

- The radar Doppler spectra are then converted from linear units $\mathrm{mm}^6\,\mathrm{m}^{-3}$ into unit of dBZ via

$$S^{\mathrm{dBZ}}(v_D) = 10\log_{10}\left(S^{\mathrm{lin}}(v_D)\right), \quad v_D \in [-v_{\mathrm{Nyq}}, +v_{\mathrm{Nyq}}] \tag{1}$$

  where $S(v_D)$ is the spectral power as a function of velocity $v_D$.

- The radar Doppler spectra are normalized by

$$\hat{S}(v_D) = \frac{S^{\mathrm{dBZ}}(v_D) - S_{\min}}{S_{\max} - S_{\min}}, \tag{2}$$

  where $S_{\max} = 20\,\mathrm{dBZ}$ and $S_{\min} = -50\,\mathrm{dBZ}$ are the maximum and minimum expected values. $S(v_D)$ values above and
below this range are set to the corresponding $S_{\max}$ and $S_{\min}$, respectively.

- Successively recorded spectra for each range gate are combined to form a time-spectrogram $\hat{S}(v_D, t)$ with $t \in [t_i - 15, t_i + 15]\,\mathrm{s}$ and $t_i$ being a time step on the temporal domain of the Cloudnet products. The grid size of Cloudnet is used as target, i.e. 30 s temporal resolution and range resolution between $30 - 45\,\mathrm{m}$, depending on the respective chirp.

- The corresponding labels are assigned: *cloud droplets present* (i.e. Cloudnet classes: liquid / mixed-phase clouds) and *no*
*cloud droplets present* (i.e. Cloudnet classes: ice / insects / drizzle / rain). Finally, a list of time-spectrograms $X$ and their corresponding labels $y$ is generated.

Figure 1 shows an exemplary feature-sample (time-spectrogram), where a fast-falling population of large ice crystals is visible as a peak at $-3\,\mathrm{m\,s}^{-1}$ and high spectral reflectivity of up to $-8\,\mathrm{dBZ}$. The second peak at $\approx 0\,\mathrm{m\,s}^{-1}$ indicates a population of supercooled liquid cloud droplets ($T = -10\,^\circ\mathrm{C}$) with reflectivity values up to $-18\,\mathrm{dBZ}$.

## 3.2 Cloudnet target classification as reference label

Supervised deep learning approaches require large amounts of pairs of input (features) and output (labels) to learn from. For this work, the Cloudnet target classification provided by the Cloudnet processing toolbox (Tukiainen et al., 2020) is used to generate the reference label. In the first step radar moments, (i.e. $Z_e$, $\bar{v}_D$, $\sigma_w$, and LDR) are calculated from the recorded Doppler radar spectra. Radar moments together with ceilometer $\beta_{\mathrm{att}}$, MWR-LWP, precipitation rate and meteorological data
are processed using the Cloudnet algorithm to derive an a-priori hydrometeor target classification. Cloudnet provides two bit masks, the category-bit, containing information on the nature of the targets for each data point (i.e. *droplets present*, *is falling*, *insects*, etc.) and the quality-bit, which contains information on the instrument detection status (i.e. *echo detected by radar*, *echo detected by lidar*, etc.). The combination of active bits yield the Cloudnet target classification and detection status (see Fig. 2). The Cloudnet liquid droplet detection investigates the shape of the ceilometer attenuated backscatter coefficient profile
$\beta_{\mathrm{att}}$ and the attenuation height above the liquid layer base (Tuononen et al., 2019). At an approximate optical thickness of





$\tau > 3$ the lidar is completely attenuated such that hydrometeor thermodynamic phase information above the attenuation height is unreliable. The top panel of Fig. 2 displays such attenuation effects in the mixed-phase layer within a convective system from 04:00 UTC to 06:30 UTC. The lidar attenuation height of 250 m, is indicated by the detection status *Radar uncorrected for liquid attenuation* in Fig. 2 (bottom). Thus, the training data set is automatically selected by unifying all data points where liquid cloud droplets are present (Cloudnet liquid category bit) and where the detection status indicates both good radar and lidar echos. This technique avoids manual labeling. By default, Cloudnet corrects the lidar-detected liquid cloud depth using radar data, by extending a liquid layer to cloud top if the detected cloud top by radar was less than 500 m above the liquid layer base, even though no clear sign of liquid is given due to the attenuation of the lidar signal. Here, to minimize the amount of falsely classified liquid-containing data points, this liquid extension to cloud top was disabled.

## 3.3 Architecture of the machine learning model

The following section introduces the machine learning model used to relate Doppler spectra morphologies to the presence of liquid cloud droplets. The output of VOODOO is a probability distribution over a discrete set of two classes, i.e. 'cloud droplets present', and 'other targets'. The machine learning approach utilizes ideas from computer vision by means of image classification via a CNN (LeCun et al., 1989; Krizhevsky et al., 2012). These methods learn the complex structure in large data sets by using optimization strategies such as Gradient Decent variants (Ruder, 2016) in combination with Backpropagation (Kelley, 1960; Hecht-Nielsen, 1989) for optimizing the internal parameters. The aim is to find a set of parameters that minimizes the error for predictions.

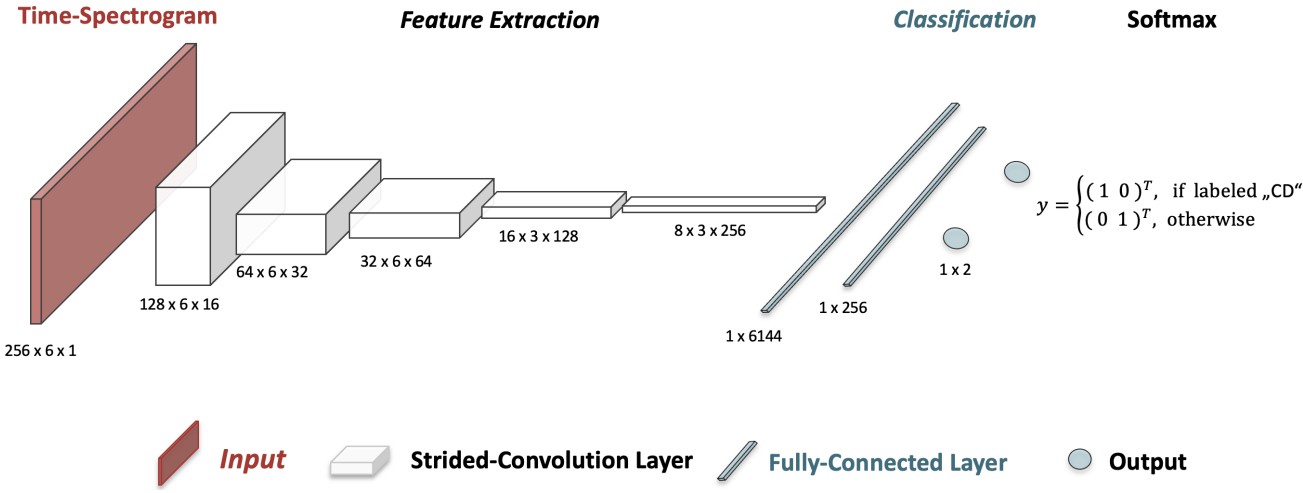

**Figure 3.** Illustration of the CNN architecture. The time-spectrogram serves as input to the VOODOO model. Five feature extraction layers follow the input layer performing strided-convolution operations. While the number of features which can be extracted increases from 16 to 256, the size of the features maps shrink from 128 by 6 to 8 by 3 pixel. The feature extraction is followed by two fully-connected layers with $8 \cdot 3 \cdot 256 = 6144$ and 256 nodes. The output is a vector with two elements, with $|y| = 1$.



A CNN classifier, implemented in PyTorch (Paszke et al., 2019), is trained on cloud radar Doppler spectra morphologies to relate to the availability of liquid cloud droplets. Fig. 3 shows the architecture of the VOODOO retrieval, which is split into
a feature extraction segment consisting of multiple convolutional layers and a classification segment consisting of two fully-connected layers. Information such as signal intensity, shape, temporal evolution, location of peaks and other morphological features of Doppler spectra can be extracted by convolution layers. CNNs (LeCun et al., 1989; Goodfellow et al., 2014; LeCun et al., 2015; Goodfellow et al., 2016) are a specialized kind of neural network for processing data that has a known, grid-like topology like a 2D grid of data points, i.e. the time-spectrogram. Using the terminology of Goodfellow et al. (2016), p. 330,
each of the convolution layers in Fig. 3 are comprised of multiple stages: stride-convolution (affine transformation) and detector stage (non-linear activation function). The 2D convolution operation extracts local features by convolving trainable 2D filters (kernels) with the input, along the velocity-time-axis (x-y-axis). During the convolution operation the kernels are shifted by two data points along the velocity axis (x-axis), called stride (Springenberg et al., 2015), to merge semantically similar features into one, hence reducing the size of the previous layer by a factor of two in each layer. Additionally, the third convolutional
layer shifts the feature map by two data points along the time axis (y-axis). As the number of extractable features increases from 16 to 256 per layer, the spatial dimension decreases from (256, 6) to (8, 3), thus the precise location of features get less relevant in deeper layers. The last convolutional layer is followed by two fully-connected layers which compute non-linear input–output mappings from the extracted 2D features. To add non-linearity to the CNN model, the linear transformation (matrix-vector-product) in each layer is applied to a non-linear activation function. The *exponential linear unit* ELU used for
all hidden layer is a smooth continuous function and easy to differentiate. ELU is defined as follows:

$$\mathrm{ELU}(x) = \begin{cases} x, & x \geq 0, \\ \alpha(\exp(x) - 1), & \mathrm{otherwise} \end{cases} \tag{3}$$

with default $\alpha = 1.0$ and $x \in [-1, 1]$. The softmax function in the last layer provides a probability for the prediction of the discrete set of two classes:

$$L = \mathrm{softmax}(z_j) = \exp(z_j) / \sum_{k=1}^{2} \exp(z_k), \ j = 1, 2 \tag{4}$$

Finally, the threshold $p^*$ controls the classification into *'cloud droplets present'* (CD) if $L > p^*$ , and *'cloud droplets not present'* (noCD) if $L \leq p^*$, respectively. This parameter $p^*$ can be manually adjusted by the user or computed automatically by receiver operating characteristics, i.e. ROC-curve analysis (Zou et al., 2007). Note that the output of the network should be not interpreted as probability density function, which represents a notion of confidence, instead it is a pseudo probability or likelihood for each class. For better readability the term "pseudo" is omitted subsequently.

## 3.4 Training and validation set

The presented machine leaning technique is trained on 10% of DACAPO-PESO (Punta Arenas) data measured by the RPG-FMCW-94-DP Doppler cloud radar. While larger training sets usually increase the performance in deep learning models, no



major advantages could be observed, when more than 10 % of the available data set was used for training. The remaining 90% of data from Punta Arenas and 100% of data obtained in Leipzig, are used to validate the predictive performance. For Leipzig, a list of 50 days was removed from the entire data set, by filtering out days with clear-sky only, the sole presence of very thin clouds with LWP below $20\,\mathrm{g\,m^{-2}}$, precipitation lasting all-day, artifacts in the radar spectra from a nearby construction crane, and ceilometer observations being fully attenuated below $200\,\mathrm{m}$. Isolated data points (speckles) are removed from the data set. Finally, the outer 3 data points along the edge of each observed cloud were omitted from the training and validation data set, to reduce the effects of partial beam filling (radar sample volume partially filled with atmospheric targets).

### 3.5 Training process

During training, VOODOO is fed by a time-spectrogram and outputs a vector of scores, one for each category. An objective function measures the error (or loss) between the output and the desired target. Via a modified stochastic gradient descent the categorical cross-entropy loss function:

$$J = -\sum_j y_j \log q_j, \qquad j = 1, 2,$$ (5)

is minimized, where $q_j = \mathrm{softmax}(z_j) \in [0, 1]$ and $q_0 + q_1 = 1$ is the predicted pseudo-probability for class $j$, and $y_j \in \{0, 1\}$ where $y_0 + y_1 = 1$, representing the one-hot-encoded labels (see Sec. 3.1). One-hot encoding is used to convert the categorical data (labels) to numerical data required by machine learning algorithms. To reduce the loss function $J$, the internal hyper-parameters (kernels and fully-connected layer weights) are adjusted by applying the adaptive moment estimation (Adam) optimization method (Kingma and Ba, 2017), which uses a stochastic gradient-based optimization approach. Gradient decent (Ruder, 2016) in combination with backpropagation (Kelley, 1960; Hecht-Nielsen, 1989) adjusts the hyperparameters itera-tively to minimize $J$. Processing was done on a GPU workstation using four NVIDIA RTX 8000, training 10 epochs takes approximately 15 min. However, the optimization plateaus already after 3 epochs. The term epoch refers to one cycle through the full training data set and can also be considered as the iteration of optimization.

### 3.6 Post-processing

In the last step, the raw VOODOO predictions $L$ (Eq. 4) are assigned to the original coordinates in the spatio-temporal domain and convolved with the 2-dimensional Gaussian filter

$$G(x, y) = \frac{1}{2\pi\sigma^2} \exp\left(-\frac{x^2 + y^2}{2\sigma^2}\right)$$ (6)

where $x$ and $y$ corresponds to the time and range indices, respectively and $\sigma = 1$, to generate more coherent structures. After classification into *'no cloud droplets present'* and *'cloud droplets present'* using the $p^*$ threshold (see Sec. 3.3) all data points with mean Doppler velocity below $-3\,\mathrm{m\,s^{-1}}$ are re-classified as *'no cloud droplets present'*. This value gives a good compromise between physically reasonable (drizzle/rain correspond to faster-falling hydrometeors) and the influence of gravity waves, the latter being omni-present in Punta Arenas (Radenz et al., 2021).





## 3.7 Performance validation

This section lists common measures for validation of the predictive performance of VOODOO. In the following, "CD" refers to
'cloud droplets present' samples (i.e. the positive class) and "noCD" to 'cloud droplets not present' samples (i.e. the negative
class), respectively. The confusion matrix $\mathcal{C}(p^*)$ summarizes all counts of the correct and misclassified data samples for each
class. This 2-by-2 matrix consists of the numbers of correctly identified CD (true positives, TP, hits) and noCD (true negative,
TN) samples on the main-diagonal of the matrix, as well as falsely classified noCD (false positive, FP, false alarm) and CD
(false negative, FN, misses) samples on the off-diagonal respectively, i.e.

$$\mathcal{C}(p^*) = \begin{pmatrix} \text{TP} & \text{FP} \\ \text{FN} & \text{TN} \end{pmatrix}. \tag{7}$$

The value of $p^*$, i.e. the probability threshold necessary for classification as CD (see Sec. 3.3), is adjustable and controls the
ratio of false positive and false negative predictions. Increasing $p^*$ increases the amount of false negatives and decreases the
number of false positives. Vice-versa is true, if $p^*$ is decreased. The thresholds were selected manually after investigating the
ROC-curve (Zou et al., 2007), yielding good balance between misses (FN) and false alarms (FP). The listing below gives the
performance scores used to evaluate the retrieval, similar to Kalesse-Los et al. (2022).

1. precision or positive predictive value (PPV): A real value between 0 and 1, where 1 is the perfect score.

$$\text{precision} = \frac{\text{TP}}{\text{TP} + \text{FP}}, \tag{8}$$

    i.e. the fraction of how many predictions where correctly classified as CD (i.e. TP) and the sum of TP and predictions
    falsely classified as CD (i.e. FP). In the context of this work, it measures the amount of CD overestimation.

2. negative predictive value (NPV): A real value between 0 and 1, where 1 is the perfect score.

$$\text{NPV} = \frac{\text{TN}}{\text{TN} + \text{FN}}, \tag{9}$$

    i.e. the fraction of how many noCD predictions where correctly classified as such (i.e. TN) and the sum of TN and
    predictions falsely classified as noCD (i.e. FN). In the context of this work, it measures the amount of correctly identified
    noCD samples.

3. recall or true positive rate (TPR): A real value between 0 and 1, where 1 is the perfect score.

$$\text{recall} = \frac{\text{TP}}{\text{TP} + \text{FN}}, \tag{10}$$

    i.e. the fraction of TP and the sum of TP and liquid-containing pixels, which were falsely classified as noCD (i.e. FN).
    In the context of this work, recall measures the amount of CD underestimation. The closer recall gets to 1, the less likely
    is missing actual CD.





4. selectivity or true negative rate (TNR): A real value between 0 and 1, where 1 is the perfect score.

$$\text{selectivity} = \frac{\text{TN}}{\text{TN} + \text{FP}},\tag{11}$$

i.e. the fraction of TN and the sum of TN and noCD samples, which were falsely classified as CD (i.e. FP). As selectivity approaches 1, the number of false alarms (FP) approaches 0.

5. accuracy: A real value between 0 and 1, where 1 is the perfect score.

$$\text{accuracy} = \frac{\text{TP} + \text{TN}}{\text{TP} + \text{TN} + \text{FP} + \text{FN}}\tag{12}$$

i.e. the fraction of all correct predicted CD pixels and the sum of all samples. In the context of this work it measures the overall fraction of correct versus incorrect predictions, e.g. accuracy $= 0.75$ if the retrieval correctly classifies 3 out of 4 input samples.

6. correlation coefficient $r^2_{\text{LLT}}$: Correlation between MWR-LWP and retrieved liquid layer thickness (LLT), where LLT is the geometric extent of all liquid layers above the instrument, i.e. the geometrical depth of the retrieved liquid mask introduced by Luke et al. (2010). All time series were smoothed with a box-window of 10 min.

7. correlation coefficient $r^2_{\text{LWP}}$: Correlation between MWR-LWP and retrieved adiabatic liquid water path $\text{LWP}_{\text{ad}}$: The MWR-LWP time-series is correlated with the retrieved adiabatic LWP time-series $\text{LWP}_{\text{ad}}$, computed from the spatio-temporal CD mask, temperature and pressure profiles from ECMWF using an adiabatic cloud parcel model introduced by Karstens et al. (1994) for better physical interpretation. All time series were smoothed with a box-window of 10 min.

8. The influence of LWP and atmospheric turbulence on the performance scores is investigated using the MWR and an estimation of the rate at which turbulence kinetic energy is transferred from larger eddies into smaller ones and eventually dissolves into thermal energy, called eddy dissipation rate $\varepsilon_{\text{DR}}$. The derivation of the method estimating the $\varepsilon_{\text{DR}}$ from Cloudnet horizontal and vertical wind speeds and radar mean Doppler velocity was introduced by Borque et al. (2016). The computation is done by an implementation developed by Griesche et al. (2020).

9. Appendix B shows similarities in the distribution of predictions and the ground-truth via the probability density function (PDF) of TP, FP, FN, TN, CD, and noCD. The distributions of six variables from radar and lidar observations $(Z_e, v_D, \beta_{\text{att}}, \varepsilon_{\text{DR}}, \text{LDR}, T)$ are investigated.

## 4  Results and discussion

In this section, the predictive performance of the presented retrieval is investigated. First, in Sections 4.1 and A, detailed analyses of two case studies from Punta Arenas and Leipzig are presented. Secondly, statistics for the entire 18 month long data set are presented in Section 4.2. Table 4 shows an overview of achieved performance scores and correlation coefficients both for case studies and statistics.





**Table 4.** Binary classification performance metrics and correlation coefficients with respect to MWR-LWP and adiabatic LWP from cloud liquid mask. Two case studies and the statistics from Punta Arenas and Leipzig are shown. The abbreviations **C** and **S** refer to a case studies and statistics for the full data set, respectively.

|  | $p^*$ | precision | NPV | recall | selectivity | accuracy | $r^2_{\text{LLT}}$ VOODOO / Cloudnet | $r^2_{\text{LWP}}$ VOODOO / Cloudnet |
|---|---|---|---|---|---|---|---|---|
| Punta Arenas **C** | 0.4 | 0.86 | 0.90 | 0.65 | 0.96 | 0.89 | 0.79 / 0.37 | 0.78 / 0.49 |
| Leipzig **C** | 0.3 | 0.91 | 0.67 | 0.32 | 0.98 | 0.70 | 0.80 / 0.48 | 0.76 / 0.47 |
| Punta Arenas **S** | 0.4 | 0.60 | 0.75 | 0.20 | 0.96 | 0.77 | 0.48 / 0.20 | 0.45 / 0.18 |
| Leipzig **S** | 0.3 | 0.64 | 0.75 | 0.31 | 0.96 | 0.73 | 0.50 / 0.24 | 0.48 / 0.22 |

## 4.1 Case study Punta Arenas

Fig. 4 shows the observations from Aug. 1, 2019 in Punta Arenas from radar and lidar perspective. From 01:30 UTC to 04:00 UTC, a mid-level stratiform cloud is present, which begins to form light precipitation about 30 min after observation onset. High $\beta_{\text{att}}$ (Fig. 4 D) indicates a liquid cloud-top at about $-15\,°C$ in 3 km altitude, producing low amounts of small precipitating ice particles indicated by low $\beta_{\text{att}} < 10^{-7}\,\text{m}^{-1}\,\text{sr}^{-1}$ below the liquid layer. After 05:00 UTC, a multi-layer mixed-phase cloud was observed, indicated by high $\beta_{\text{att}}$ (Fig. 4 D), with liquid cloud base heights (LCBH) of 1.3 km and 2.8 km,

respectively. From 06:00 UTC, the upper LCBH dropped to 2 km, then starts to rise again. At 07:15 UTC the liquid layer continues to increase in altitude from 2.0 – 2.6 km at 08:45 UTC. During this time the reflectivity values ranging between $-27$ to $+10$ dBZ between 2.0 – 2.4 km altitude indicate a population of larger ice particles. Several smaller liquid-dominated clouds were observed by the ceilometer around 04:45 UTC at 4.7 km with temperatures $T < -25\,°C$, and after 05:30 UTC at 1.2 km ($T < -5\,°C$), and 08:30 UTC at 0.5 km height ($T < 0\,°C$), where only some data points exceed the minimum detection

capabilities of the cloud radar.

Figure 5 (A) shows the output of VOODOO with threshold $p^* = 0.4$ (see Tab. 4), which provides a good compromise between FP and FN predictions. Light gray cells ($L \leq 0.4$) indicate noCD volumes and $L > 0.4$ CD bearing, respectively. A visual comparison between bands of high $\beta_{\text{att}}$ in (Fig. 4 D) and predicted CD (Fig. 5 A) show good temporal and spatial agreement, as indicated by the cloud base plotted as red dots. Figure 5 (B) shows a visual reference of CD false-alarms marked

by FP and CD misses marked by FN, respectively. With an accuracy $= 0.890$, almost 9 of 10 data points were correctly classified. An individual look on precision $= 0.86$ shows a false alarm rate ($1-$ precision) of only 14%, while a recall $= 0.65$ was achieved. Visible in Fig. 5 (B) are TP (light blue) and TN (light gray), where the VOODOO predictions match the Cloudnet classification. In contrast, larger clusters of FN data points (yellow) occur mostly at liquid cloud base (2:00 – 4:00 UTC, at 2.7 km) and in thin pure liquid clouds (5:00 – 6:00 at 1.3 km), thus reducing the recall value. Those FN

predictions, which are responsible for the deviation of the recall from 1, are expected due to the lower sensitivity of the

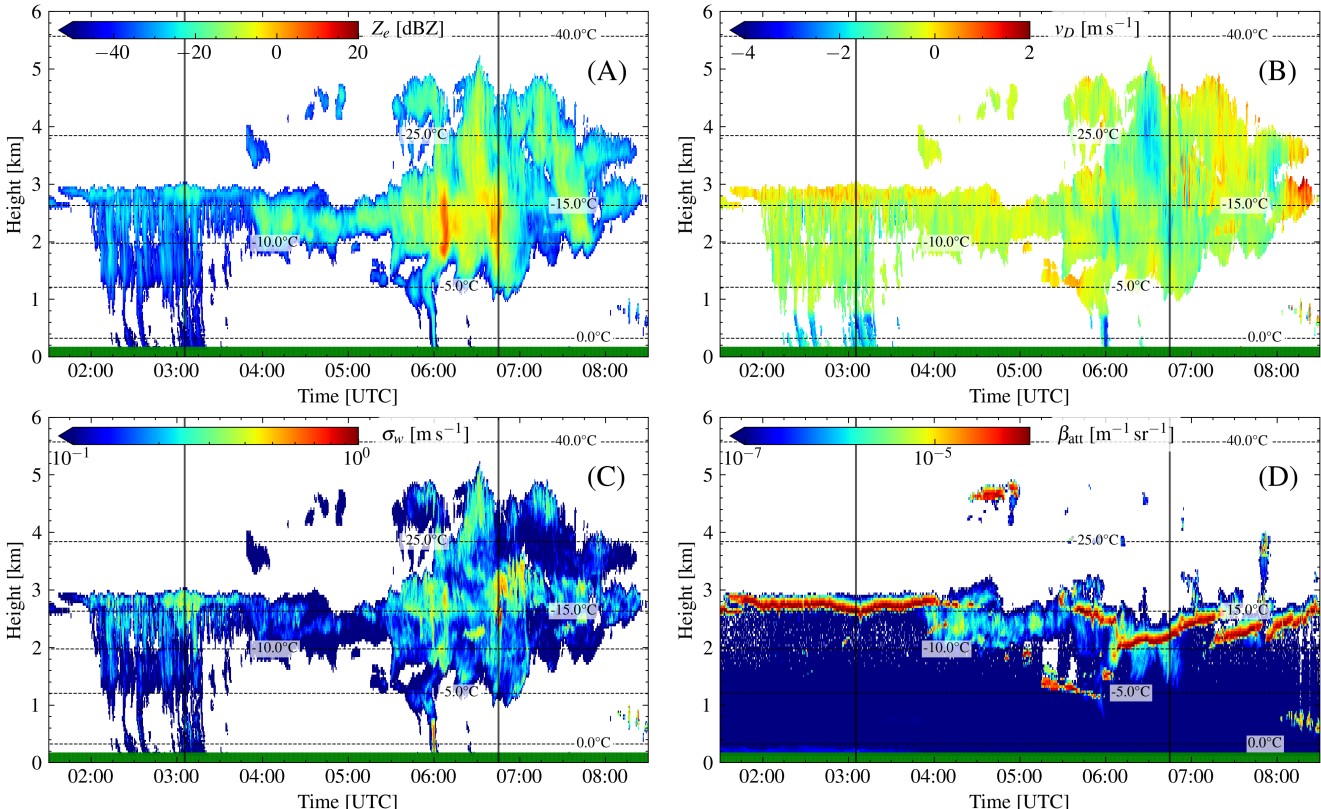

**Figure 4.** Case study of Aug. 1, 2019 in Punta Arenas, Chile. (A) radar reflectivity factor $Z_e$, (B) radar mean Doppler velocity $\bar{v}_D$, (C) radar spectrum width $\sigma_w$, (D) ceilometer attenuated backscatter $\beta_{att}$. Dashed lines depict the isotherm lines from ECMWF temperature profiles. The green horizontal line at $y-$axis$= 0$ indicates no rain was measured at ground. Solid vertical lines mark locations of the range-spectrograms, shown in Fig.7.

radar to liquid droplets compared to the lidar. Another cause of the FN predictions could be that liquid-only clouds or cloud volumes with low amounts of liquid droplets produce less Doppler spectra features, e.g. single peaks with low intensity or below noise floor or by superimposing the liquid peak on the much larger ice peak, which makes the features of the liquid peak disappear (i.e. non-separable from the ice). On the other hand, smaller clusters of FP predictions (red) occur below the

ceilometer cloud base height (CBH). Those misclassifications are possibly caused by a higher spectrum width likely due to atmospheric turbulence. At the same time it should be noted that also Cloudnet's classification cannot be perfect even though it is used as ground truth here. This limits the achievable maximum of the quality metrics used in this study.

Note that there are approximately twice as many valid noCD than CD samples available in both data sets. This imbalance of the validation set makes the interpretation of performance metrics more difficult. Therefore, performance of VOODOO is

validated on multiple binary classification metrics and independent observations from MWR. The LLT, as well as $LWP_{ad}$ (cal-

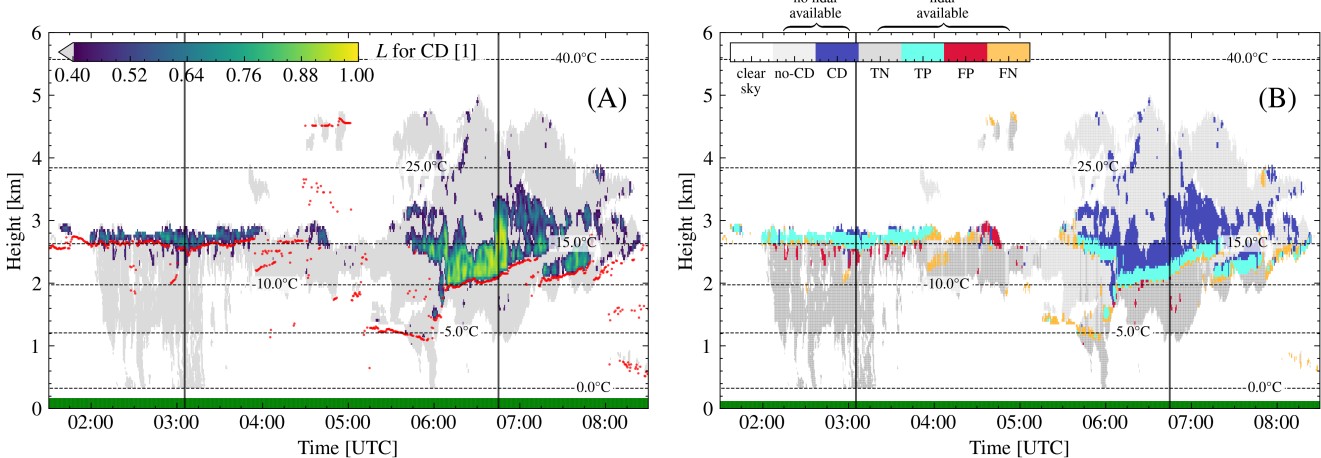

**Figure 5.** Probability for the presence of cloud droplets for case study of Aug. 1, 2019 in Punta Arenas, Chile. (A) VOODOO output: probability for CD, (B) VOODOO prediction status. Dashed lines depict the isotherm lines from ECMWF temperature profiles. Red dots in (A) indicate the first ceilometer CBH. The green horizontal line at $y-$axis$=0$ indicates no rain was measured at ground. Solid vertical lines mark locations of the range-spectrograms, shown in Fig.7.

culated according to Karstens et al. (1994)) both correlate remarkably well with the measured MWR-LWP (see Fig. 6 A and B) reaching values for LLT (0.79) and LWP (0.78). In contrast, Cloudnet achieves significantly lower correlation coefficients with respect to LLT (0.37) and LWP (0.49). The geometric extent of liquid water layers retrieved with Cloudnet is only meaningful for optically thin and single layer mixed-phase clouds, since the attenuated ceilometer signal (i.e. 06:00 - 08:00 UTC) cannot

cover the complete liquid CD distribution in the atmosphere beyond lidar attenuation, thus underestimating the thickness of deep liquid-containing layers (see: black line in Fig. 6 A).

To illustrate the performance of VOODOO better, two range-spectrograms, $\beta_\text{att}$ and CD pseudo-probability plots are shown in Fig. 7. In Fig. 7A a bi-modal distribution is observed at altitudes between $2.6 - 2.9\,\text{km}$ along with high ceilometer $\beta_\text{att} \approx 10^{-4}\,\text{m}^{-1}\,\text{sr}^{-1}$ indicating a population of liquid droplets near cloud-top and matching the prediction of VOODOO

with Cloudnet. Below 2.6 km, smaller ice crystals are falling out of the mixed-phase cloud top, which are melting and form drizzle drops at approximately 1 km altitude. Fig. 7B shows the range-spectrogram at 06:45:00 UTC. Cloud top is detected at 4.3 km, showing a mono-modal distribution at Doppler velocities of $-1.5$ to $-0.5\,\text{m}\,\text{s}^{-1}$. At 3.3 km altitude, the spectrum width suddenly increases rapidly showing a skewed distribution with $\bar{v}_D$ between $-1.0$ to $0.0\,\text{m}\,\text{s}^{-1}$. As the altitude decreases, the spectrum splits into a clearly separable bi-modal distribution at altitudes below 3.0 km, indicating a CD population at $0\,\text{m}\,\text{s}^{-1}$

Doppler velocity. The ceilometer shows a peak with high $\beta_\text{att} \approx 10^{-4}\,\text{m}^{-1}\,\text{sr}^{-1}$ at 2.2 km, matching the location of the bi-modality in the Doppler spectrum at the liquid layer base. Beyond an altitude of 2.4 km, the ceilometer is getting completely attenuated, such that Cloudnet's liquid droplet retrieval is not reliable anymore. In contrast, VOODOO is able to predict the



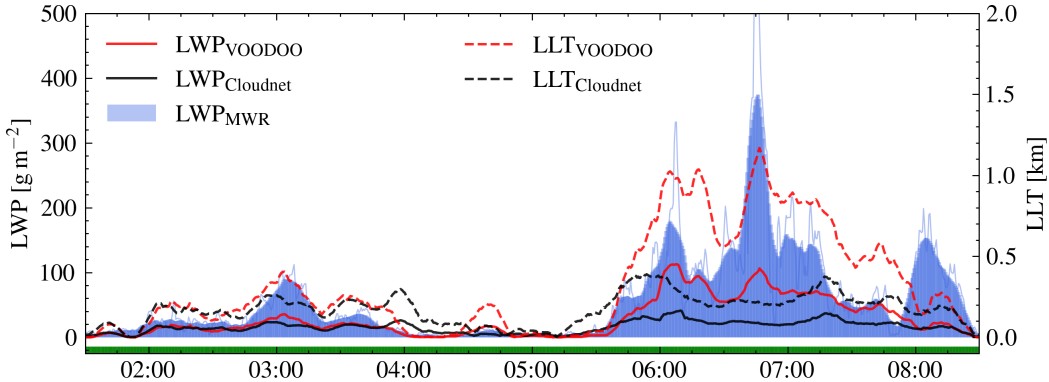

**Figure 6.** Comparison of liquid water path (LWP) and liquid layer thickness (LLT) for the case study of Aug. 1, 2019 in Punta Arenas, Chile. LWP (left y-axis, solid lines) and LLT (right y-axis, dashed lines). The thin blue line corresponds to original MWR-LWP time resolution, thick lines to 10 min smoothed data.

entire range of the liquid layer from base (2.1 km) to top (3.1 km) liquid layer. Below, only one peak with increasing $\bar{v}_D$ is visible in the spectrogram, which indicates that the larger ice crystals at higher Doppler velocities evaporate while precipitating out of the mixed-phase layer. A second case based on observations in Leipzig, Germany is presented in Appendix A.

## 4.2 Statistical analysis of the performance of VOODOO

The statistical analysis is carried out by applying VOODOO to 18 months of observations from the two different geographical sites, excluding $-10\%$ training data from Punta Arenas. This section discusses the ability of VOODOO to infer to the presence of CD from Doppler radar spectrum features for a large data set spanning over 1.5 years, which has not been done in previous studies. The Punta Arenas data set (PA) contains 220 days of observations (23 Mio data points), while the data from Leipzig (LE) counts 342 days (30 Mio data points). Fig. 8 shows the total numbers of validatable (lidar available) and non-validatable (no lidar available) data points. The numbers of negative samples TN+FP and noCD (i.e. ice, drizzle or rain) are by far the most represented classes. The ratio of TN+FP to TP+FN (i.e. the ratio of validable ice and liquid samples) is approximately 5:1 (PA) and 3:1 (LE), whereas the ratio of noCD to CD (i.e. the ratio of non-validable ice and liquid samples) is 22:1 (PA) and 9:1 (LE). Our new approach predicts additional $+50\%$ CD for PA and $+100\%$ for LE beyond the lidar attenuation height. Note that the distribution is very sensitive to the $p^*$ threshold introduced in Sec. 3.3.

Thresholds of $p^*$ for PA and LE are listed in Table 4. Two individual values for $p^*$ where chosen, to keep the false positive rate (FPR $= 1-$ selectivity) below 5% and maximize the number of correct predictions (TP and TN) while minimizing false predictions (FP and FN). The last two lines of Table 4 summarize the results found for the statistical evaluation, calculated with the sum of all valid classification results. The $r^2$ columns represent the mean correlation coefficient over the entire data set.





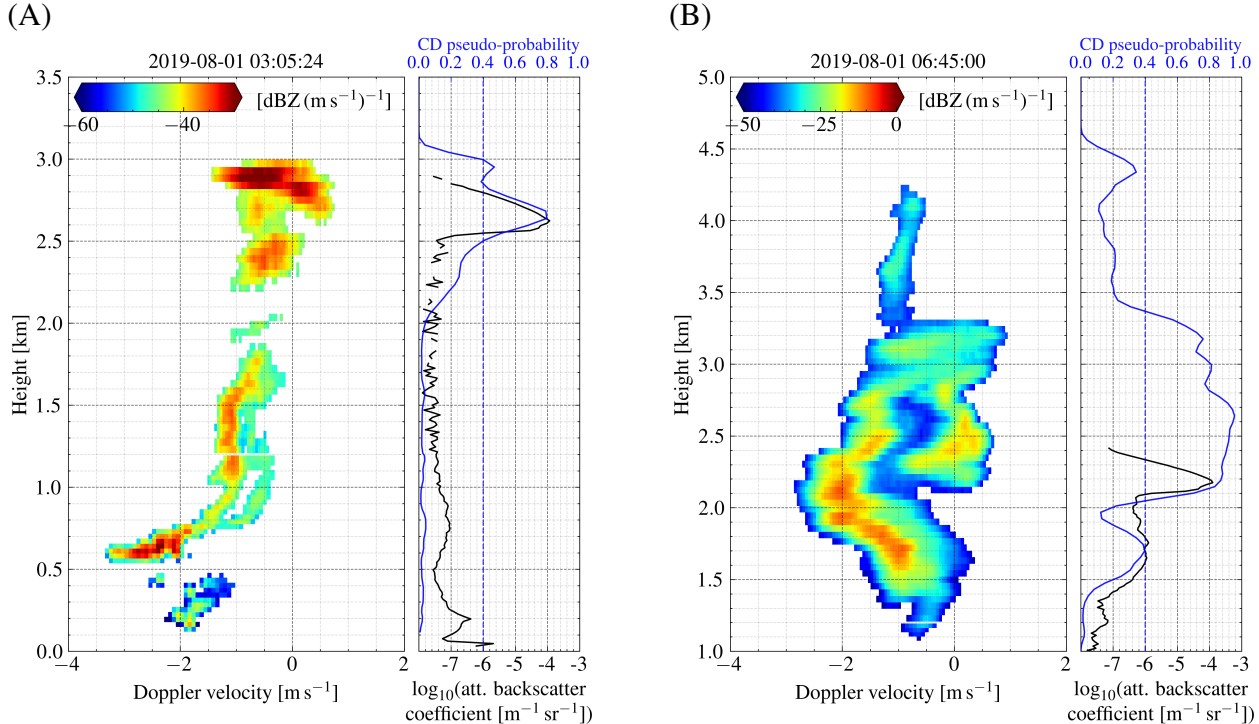

**Figure 7.** Range-spectrogram (left panel) – attenuated backscatter coefficient (right panel, solid black line, bottom x-ticks) – CD probability (right panel, solid blue line, top x-ticks) profiles of Aug. 1, 2019 in Punta Arenas, Chile. The dashed blue line highlights the decision threshold $p^*$ for the presence of cloud droplets. (A) and (B) are samples for two different points in time (see black vertical lines in Fig. 4 and 5). The range-spectrogram show bi-modal distributions (A) at $2.6 - 2.9$ km and (B) at $2.1 - 2.9$ km, coinciding in altitudes with the large peaks in the attenuated backscatter profile and matching the peaks in the predictions. The black line in the right subplot of (B) displays the attenuation of the ceilometer by total signal loss in the signal above 2.2 km altitude, while VOODOO is able to relate the bi-modal signature (left panel) to the presence of droplets above 2.2 km, as it can be seen by the matching peak with the CD probability (blue line).

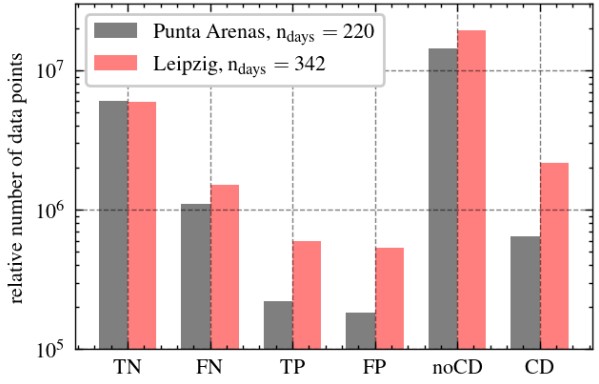

**Figure 8.** Distribution of predicted lidar-validable (TN, FN, TP, FP) and non-lidar-validable (noCD, CD) data points.

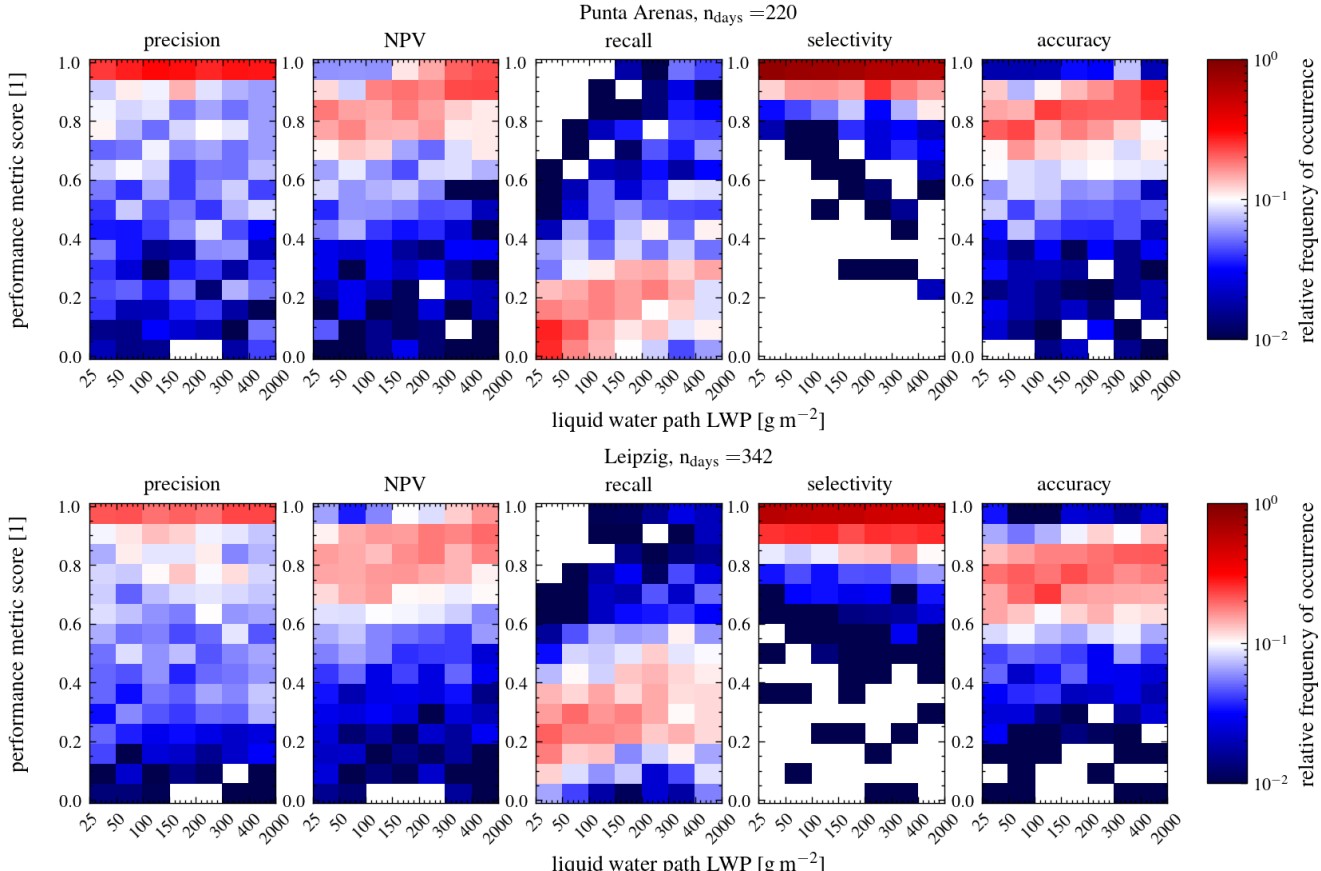

**Figure 9.** Histograms of frequency of occurrence for each performance score as function of LWP. Upper row Punta Arenas from Nov. 27, 2018 - Sep. 29, 2019 and lower row Leipzig from Dez. 16, 2020 - Mar. 6, 2022. For all scores, a value of 1 represents the perfect score.

First, the performance is analyzed by means of relative frequencies of occurrence for a specific score (i.e. precision, recall, etc.) as function of MWR-LWP. This gives an impression of how well VOODOO is able to reproduce the classification results provided by Cloudnet. The first column of Fig. 9 shows that the majority of precision score occurrences over the entire LWP range are thatight than $0.8$, with mean precision of $0.60$ (PA) and $0.64$ (LE). From this follows that VOODOO is able to
accurately relate cloud radar Doppler spectra features to CD presence, independently of LWP. The second columns of Fig. 9 shows that the majority of occurrences of the recall is below values of $0.3$. With increasing LWP, the recall score improves, but generally stays quite low. More in-depth analysis of the prediction data reveals that the majority of missed CD samples (FN), which reduce the recall score, are found close to correctly identified CD samples (TP). Low recall is obtained for the uncertainty range of the MWR-LWP (LWP $< 25\,\mathrm{g\,m^{-2}}$), where clouds contain almost no liquid water and the noCD samples
predominate 10 to 1 over CD samples. Best scores are achieved for LWP $> 100\,\mathrm{g\,m^{-2}}$. A visual reference is given by the two case studies in Fig. 5 and Fig. A2. These FN are caused by the lower sensitivity of the radar to smaller populations of liquid





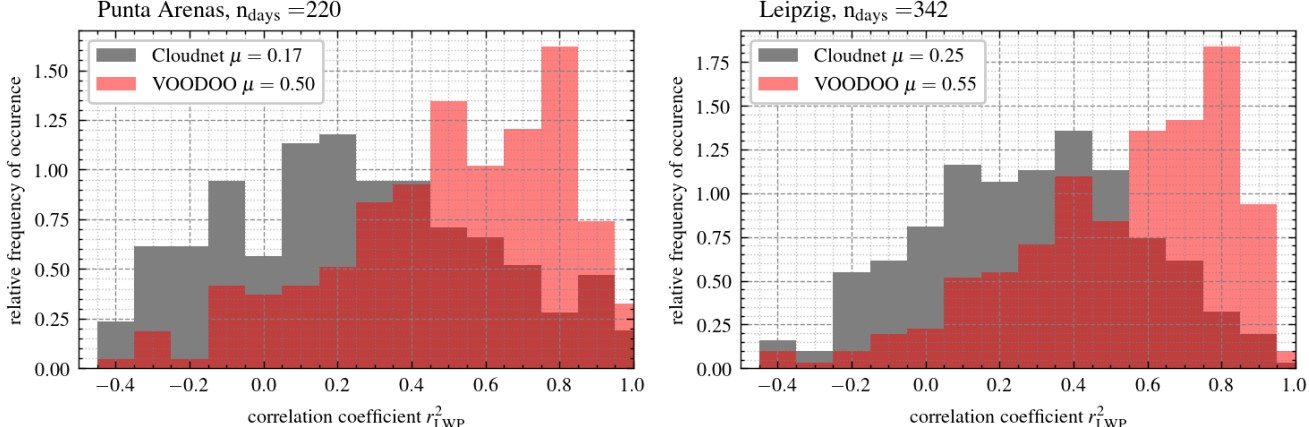

**Figure 10.** Correlation of MWR-LWP and $LWP_{ad}$, number of days with correlation coefficients achieved. Left: Punta Arenas from Nov. 27, 2018 - Sep. 29, 2019 and lower row Leipzig from Dez. 16, 2020 - Mar. 6, 2022. Median $\mu$ of $r^2_{LWP}$ given in the legend.

cloud droplets or smaller sizes of cloud droplets compared to the ceilometer. Especially thin (supercooled) liquid-only clouds observed frequently over PA, cause lower recall scores for this site. However, the linear increase in recall score with larger LWP values is more prominent in LE compared to PA, mostly due to the lower $p^*$ threshold. The accuracy values are 0.73

(LE) and 0.77 (PA). The accuracy scores (Fig. 9 column five) show most values are above 0.75, meaning 3 out of 4 samples were correctly classified, independent of LWP. Note that the LWP range below $50\,\mathrm{g\,m^{-2}}$ contains the largest absolute number of occurrences in the histogram.

The second statistical evaluation is carried out by analyzing the correlation coefficient $r^2_{LWP}$ between retrieved MWR-LWP and $LWP_{ad}$ for each day of observation. Fig. 10 shows the relative frequency of occurrence as a function of $r^2_{LWP}$ values.

Both sites display similar $r^2_{LWP}$ distributions, with the CNN-based $LWP_{ad}$ having stronger correlations with the MWR-LWP than Cloudnet-based $LWP_{ad}$. Median $r^2_{LWP}$ values $\mu$ for Cloudnet are 0.17 (PA) and 0.25 (LE), whereas VOODOO is able to improve the correlation of the LWP to 0.50 (PA) and 0.55 for (LE). Despite the limits of this validation method, this shows that VOODOO is able to approximate the values of the MWR-LWP better compared to Cloudnet because it detects cloud droplets beyond the lidar attenuation. Thus, VOODOO's improved liquid detection (amount and location) can potentially improve

higher level retrieval products such as liquid water content.

To evaluate the impact of turbulence on the VOODOO predictions, the relative frequency of performance scores as function of turbulence eddy dissipation rate $\varepsilon_{DR}$ within a range of $(-9, 0)$ in units of $\log_{10}([\mathrm{m^2\,s^{-3}}])$ are displayed in Figure 11. Note that the majority of data points (>95%) range between values of $-7 < \log_{10}(\varepsilon_{DR}) < -1$. The precision for PA and LE both show most occurrences near a score of 1. However, the frequency of NPV scores near 1 decrease slightly as $\varepsilon_{DR}$ increases,

which can be caused by reducing of the number of TN or the increase of FN samples. Since recall performance improves slightly as the $\varepsilon_{DR}$ increases, it follows that the number of TN is reduced, thus most sample volumes containing no cloud



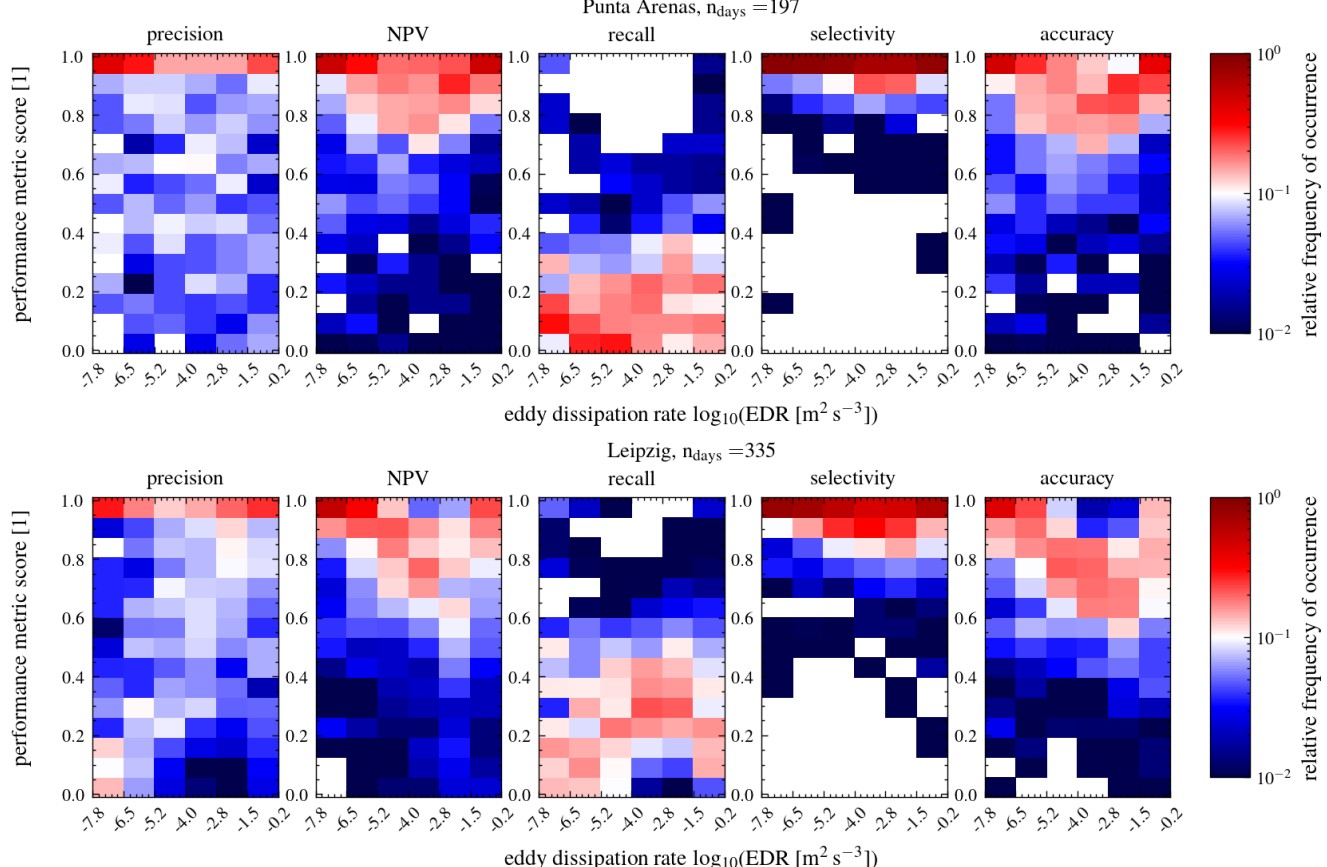

**Figure 11.** Histograms of frequency of occurrence for each performance score as function of $\varepsilon_{\mathrm{DR}}$. Upper row Punta Arenas from Nov. 27, 2018 - Sep. 29, 2019 and lower row Leipzig from Dez. 16, 2020 - Mar. 6, 2022. For all scores, a value of 1 represents the perfect score.

droplets show low $\varepsilon_{\mathrm{DR}}$. The accuracy score is mostly influenced by TN values, thus it follows the trend of the NPV. The selectivity scores, for both PA and LE shows a very narrow distribution near the perfect score 1. We can conclude that the influence of turbulence on the predictive performance of VOODOO is minor.

## 5 Summary and Outlook

The supervised machine learning retrieval VOODOO is presented, which predicts the presence of cloud droplets from cloud radar Doppler spectra. Time spectrograms are processed by VOODOO, to predict directly a probability for the presence or absence of cloud droplets. The model is trained on long-term ground-based remote-sensing observations from Punta Arenas in Chile. The a-priori ground-truth is given by the Cloudnet algorithm which is a multi-instruments retrieval that processes radar, ceilometer, MWR, and ECMWF forecast model data into higher level products (e.g. the atmospheric target classification).





The performance is validated in detail on two case studies from different geographical locations, i.e. Punta Arenas, Chile and Leipzig, Germany located in the mid-latitudes of the Southern and Northern hemisphere, respectively. This is done to test whether the trained CNN is also applicable for very different orographic conditions and aerosol loads influencing the occurrence of liquid-containing clouds and their properties over both sites. In addition to the case studies, for the first time,
long-term observations of both sites are used to investigate the retrieval's robustness to new data.

The case study shows the ability of VOODOO to extract features from radar Doppler spectra and infer the presence of CD for the desired type of mixed-phase clouds. Due to the limitation of pixel-by-pixel comparison, instrument and retrieval uncertainties, the design of the study does not allow a perfect score of 1. Nevertheless, for the long-term observations, VOODOO achieves good precision ($> 0.60$) and accuracy ($> 0.73$), confirmed indirectly by a strong correlation between MWR-based
and $LWP_{ad}$ ($> 0.63$). Due to lower sensitivity of cloud radar to small liquid droplets compared to lidar, the recall score is only ($< 0.2$). Overall, more FN predictions than FP are responsible for this, FN occurring more frequently at the lowest range gates of liquid or mixed layers, or in thin and pure liquid layers (warm or supercooled) with low LWP. Overall, VOODOO performs best for (multi-layer) stratiform, deep mixed-phase cloud situations with LWP $> 100\,\mathrm{g\,m^{-2}}$, where the number of liquid layers and the liquid layer depth often extends far more than previously assumed.

In conclusion, VOODOO performs best for (multi-layer) stratiform, deep mixed-phase cloud situations with LWP $> 100\,\mathrm{g\,m^{-2}}$. From this analysis we learn that both methods Cloudnet and VOODOO have their strengths and weaknesses. Clearly, Cloudnet's lidar-based approach has an advantage in detecting thin liquid water layers, whereas VOODOO's radar approach can be used primarily to reveal hidden liquid water layers beyond lidar attenuation. Radar observations during strong precipitation (Cloudnet rain-flag active), where no meaningful LWP can be retrieved from MWR observations suffer strongly from liquid
attenuation, thus features in cloud radar Doppler spectra are less reliable (not shown). The thin (supercooled) liquid-only clouds show less pronounced spectral features, thus are more similar to single ice crystal peaks in the Doppler spectra and are, as a result, often misclassified.

It has been demonstrated that the VOODOO method could be a powerful addition to the existing Cloudnet target classification, making the detection of liquid layers beyond complete lidar attenuation possible. Additional validation methods
are needed to better quantify the performance. These methods could include the comparisons of space-borne remote-sensing observations with predictions to add cloud-top liquid-containing data points to the validation data set, or air-borne in-situ measurements for hydrometeor target classifications and radiation measurements with radiative closure studies (Barrientos Velasco et al., 2020). Note that all of the three mentioned ideas introduce their own uncertainties.

The ability to detect liquid beyond lidar attenuation is a major step in the field of Doppler spectrum-based analysis of
vertically-pointing ground-based radar observations. The described spectrum-based hydrometeor phase partition methodology can be regarded as a modular component applicable to Doppler cloud radar spectra. A synergy of the novel approach VOODOO and Cloudnet would complement each other perfectly and is planned to be incorporated into the Cloudnet algorithm the near future.

Note that the outlined technique has been tailored to an RPG-FMCW-94-DP cloud radar, other radars, ground-based or
even space-borne (if Doppler spectra are available) may alter the quality of predictions, therefore caution is advised in the





generalization of the procedure to other Doppler radar systems. Still, the same architecture can simply be retrained on the radar Doppler spectra from another cloud radar, if a sufficiently large training data set is available.

## APPENDIX

The appendix provides a second case study from Dez. 30, 2020 at LIM in Leipzig, Germany and the probability density
functions for different parameters.

### Appendix A:  Case study Leipzig

To test whether VOODOO can be transferred to another location without re-training as suggested in Kalesse-Los et al. (2022), another example case is illustrated. Figure A1 shows observations for the second case study, which evaluates capabilities on validation data from Leipzig, Germany. Between 14:00 and 19:00 UTC on Dec. 30, 2020, a multi-layer stratiform cloud
situation was observed with cloud-top at $1.2 - 1.5\,\mathrm{km}$ ($T \approx -4\,°\mathrm{C}$) for the first and $2.5 - 2.7\,\mathrm{km}$ ($T \approx -14\,°\mathrm{C}$) for the second cloud layer, respectively. Both layers with cloud-top temperatures below $0\,°\mathrm{C}$ contain supercooled liquid water, clearly visible between $1.0 - 1.5\,\mathrm{km}$ altitude (high values of $\beta_{\mathrm{att}}$, see Fig. A1 (D)) .

   Based on the ceilometer observations, the first cloud base of a shallow low level supercooled liquid cloud ($12:30 - 14:00\,\mathrm{UTC}$ at $1.2\,\mathrm{km}$) is at $1\,\mathrm{km}$ (indicated by red dots in Fig. A2A) which is $200\,\mathrm{m}$ below the predicted LCBH. The base of the low-level
supercooled liquid cloud is increasing after $14.00\,\mathrm{UTC}$ when a second mixed-phase cloud ($14:00 - 19:00\,\mathrm{UTC}$) is observed with predicted liquid cloud-top at $2.5\,\mathrm{km}$. Further, smaller clouds at altitudes between $3.0 - 3.5\,\mathrm{km}$ and cloud-top temperatures of $-20\,°\mathrm{C}$ are predicted to be cloud droplet bearing. These higher liquid-containing layers that are predicted by VOODOO are sporadically observed by the ceilometer, when it is able to penetrate the lower liquid layer at about $2.2 - 2.4\,\mathrm{km}$ and $3.5\,\mathrm{km}$ altitude. This refers to the TP data points in Fig. A2 (B).

For this case study, only very few FP at $17:45\,\mathrm{UTC}$ (at $0.75\,\mathrm{km}$) and $18:30\,\mathrm{UTC}$ (at $2.2\,\mathrm{km}$) were predicted (see Fig. A2 B), resulting in a high precision for this case of 0.91. Predicted FN data points, approximately at CBH detected by the ceilometer (see Fig. A2 B, yellow pixel), result in a recall score of 0.32. This effect is caused by cloud edge filtering and lower radar sensitivity to liquid cloud droplets (i.e. the ceilometer being able to detect smaller numbers of CD and smaller CD). VOODOO achieves high accuracy of 0.70. Further, VOODOO displays a much better agreement with temporal evolution of LLT and LWP
compared to MWR-LWP, achieving larger $r^2_{\mathrm{LLT}}$ and $r^2_{\mathrm{LWP}}$ of 0.80 and 0.76 compared to the values for Cloudnet (0.48 and 0.47, see Tab. 4).

   Figure A4 displays the range-spectrogram - attenuated backscatter coefficient - probability profiles for two time steps, indicated by vertical black lines in Fig. A1 and A2. The liquid contribution to the bi-modal radar returns is clearly visible in Fig. A4A at cloud top between $1.1 - 1.4\,\mathrm{km}$. Below $1.1\,\mathrm{km}$, the bi-modal peak merges into one peak, resulting in a mono-
modal distribution. In the transition above $1.1\,\mathrm{km}$ VOODOO could infer the presence cloud droplets, while the more sensitive ceilometer could detect the liquid cloud base one range gate below. In contrast, below liquid cloud base all noCD predictions

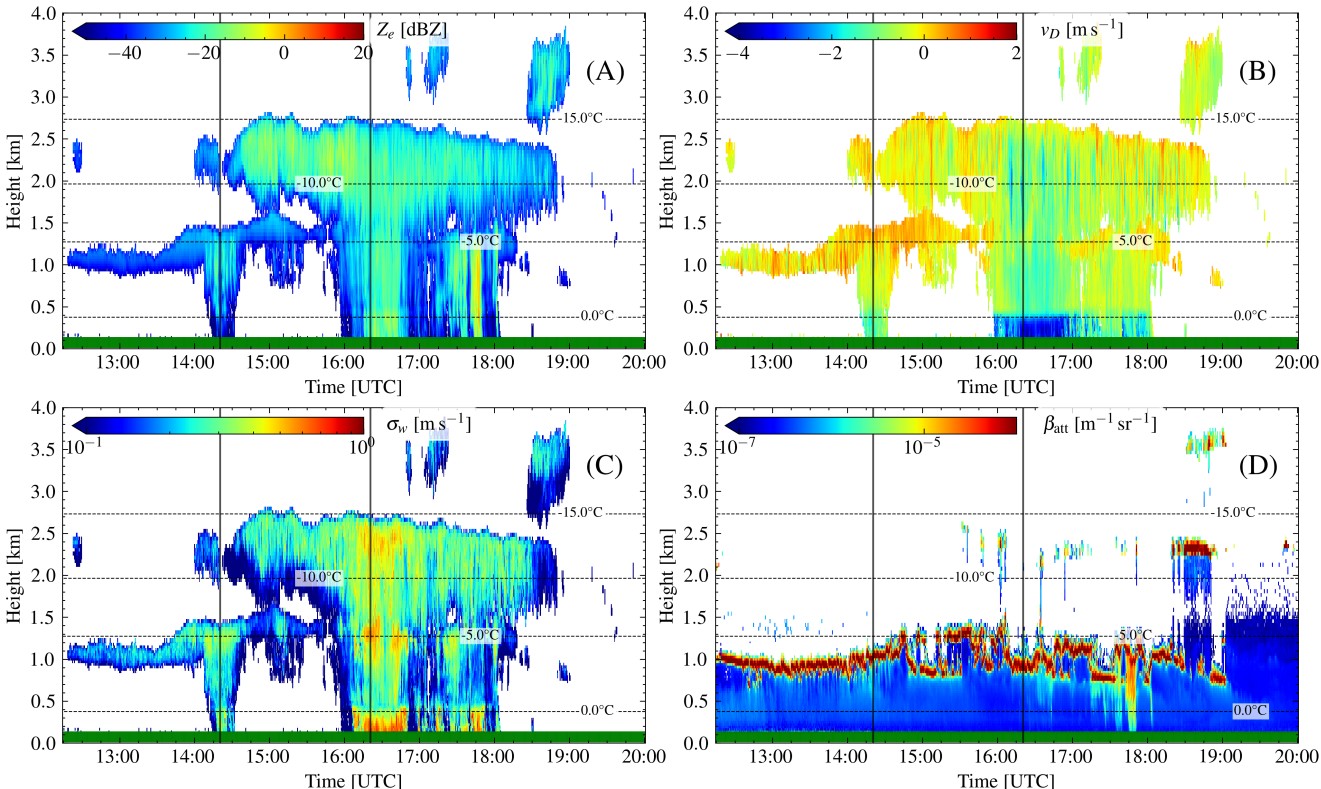

**Figure A1.** Case study of Dec. 30, 2020 in Leipzig, Germany. (A) radar reflectivity factor $Z_e$, (B) radar mean Doppler velocity $\bar{v}_D$, (C) radar spectrum width $\sigma_w$ (D) ceilometer attenuated backscatter $\beta_{\text{att}}$. Dashed lines depict the isotherm lines from ECMWF temperature profiles. The green horizontal line at $y-$axis$= 0$ indicates no rain was measured at ground. Solid vertical lines mark locations of the range-spectrograms, shown in Fig.A4.

were classified correctly by VOODOO. After 14 UTC a multi-layer cloud situation is observed and investigated through the second range-spectrogram (Fig. A4B. At cloud top (2.5 km altitude and $T = -12°$C) the spectrum shows a skewed distribution with high $\sigma_w$ and $\bar{v}_D$ at approximately $0 \, \text{m s}^{-1}$. VOODOO predicts the probability for CD $> 0.3$. At $1.0 - 1.5$ km ($T = -5°$C)
another liquid bearing layer is revealed by VOODOO, indicated by the bi-modal distribution, where the liquid contribution shifts from $0 \, \text{m s}^{-1}$ to $1 \, \text{m s}^{-1}$ within a $500 \, \text{m}$ range indicating an updraft at liquid layer top. In both profiles, VOODOO demonstrated the capability to infer the presence of CD also for regions with higher spectrum width, even without clearly separated spectral features (i.e. bi-modalities).





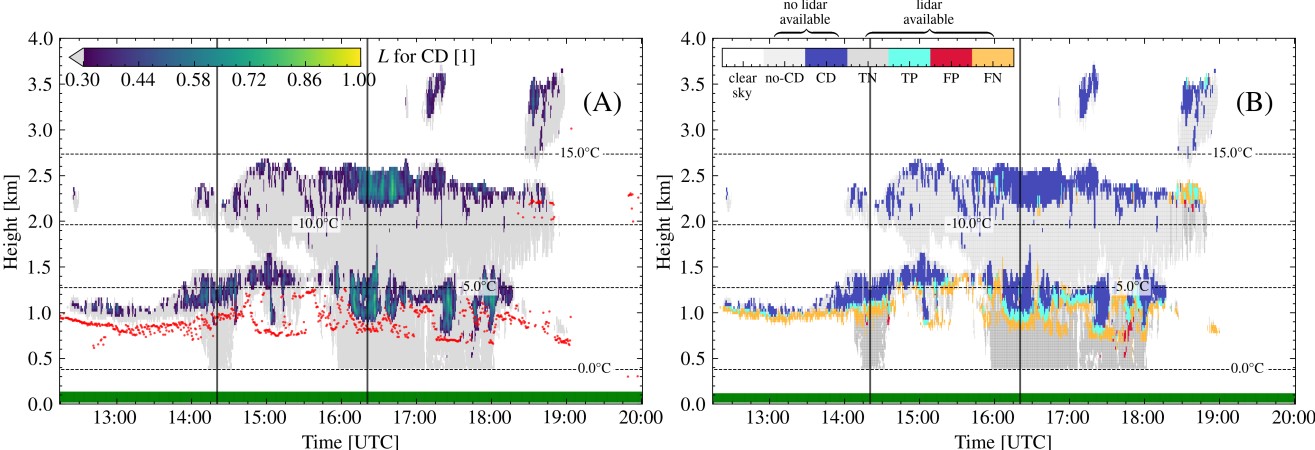

**Figure A2.** Probability for the presence of cloud droplets for case study of Dez. 30, 2020 in Leipzig, Germany. (A) VOODOO output: probability for *'cloud droplets present'*, (B) VOODOO prediction status. Dashed lines depict the isotherm lines from ECMWF temperature profiles. Red dots in (A) indicate the first ceilometer CBH. The green horizontal line at $y-$axis$= 0$ indicates no rain was measured at ground. Solid vertical lines mark locations of the range-spectrograms, shown in Fig.A4.

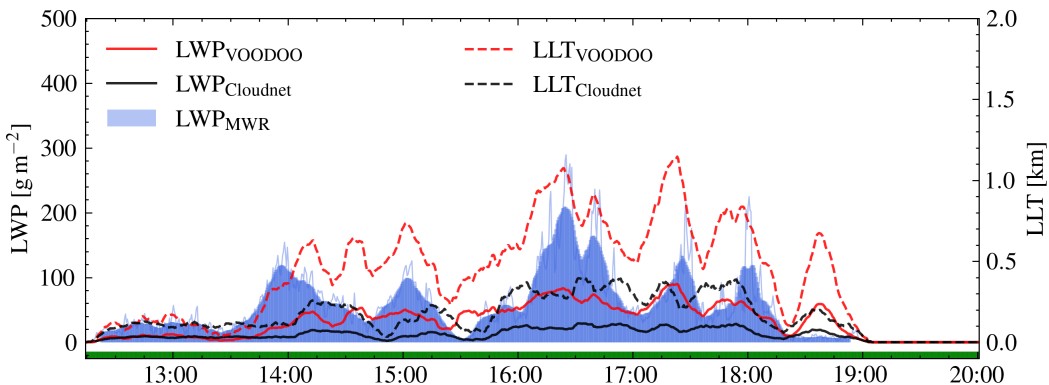

**Figure A3.** Comparison of liquid water path (LWP) and liquid layer thickness (LLT) for the case study of Dec. 30, 2020 in Leipzig, Germany. LWP (left y-axis, solid lines) and LLT (right y-axis, dashed lines). The thin blue line corresponds to original MWR-LWP time resolution, thick lines to 10 min smoothed data.





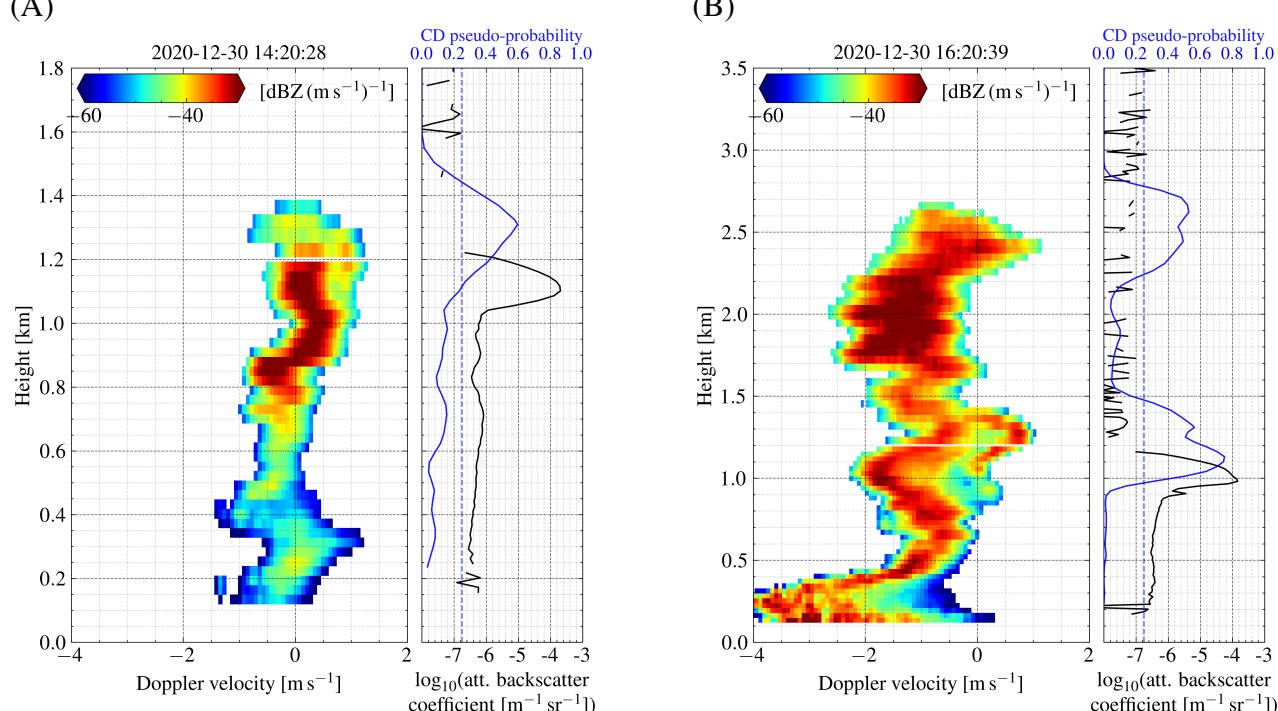

**Figure A4.** Range-spectrogram (left panel) – attenuated backscatter coefficient (right panel, solid black line, bottom x-ticks) – CD probability (right panel, solid blue line, top x-ticks) profiles of Dec. 30, 2020 in Leipzig, Germany. The dashed blue line highlights the decision threshold $p^*$ for the presence of cloud droplets. (A) and (B) are samples for two different points in time (see black vertical lines in Fig. A1 and A2). The left panel show bi-modal distributions (A) at $1.1 - 1.3$ km and (B) at $0.8 - 1.4$ km, coinciding in altitudes with the large peaks in the attenuated backscatter profile and matching the peaks in the predictions. More liquid is found by VOODOO in (B) at cloud-top ($2.3 - 2.7$ km) with enhanced spectrum width and centered near $0 \, \text{m s}^{-1}$.

## Appendix B: Probability density functions

A detailed discussion of the PDFs of different variables is listed below, given the notation $f_a^b$, with $a \in \{Z_e, v_D, \beta_{\text{att}}, \varepsilon_{\text{DR}}, \text{LDR}, T\}$ and $b \in \{\text{TP, FP, FN, TN, CD, noCD}\}$ used to distinguish individual distributions for the two Figures B1 and B2.

(A) radar reflectivity factor $Z_e$ in units of [dBZ]:

$f_{Z_e}^{\text{TP}}$ shows two distinct peaks at $-30$ dBZ for both PA and LE (i.e. liquid-only clouds) and a second peak at $-10$ dBZ for PA and $-7$ dBZ for LE (i.e. mixed-phase clouds). $f_{Z_e}^{\text{TN}}$ shows a single peak centered at approx. $-23$ dBZ for both

PA and LE with large variance. $f_{Z_e}^{\text{FP}}$ for the PA data show two separable peaks centered at $-28$ dBZ (lower amplitude) and $-2$ dBZ (larger amplitude). However, the two peaks in $f_{Z_e}^{\text{FP}}$ for LE data $-26$ dBZ (liquid-dominated mixed-phase clouds) and at $-7$ dBZ (ice-dominated mixed-phase volumes). $f_{Z_e}^{\text{FN}}$ shows a slightly skewed distribution for both LE with peaks at $-36$ dBZ (LE) and $-27$ dBZ (PA), indicating that most volumes with cloud droplets which were not correctly predicted by VOODOO have low reflectivity. Nevertheless, the LE data has a much more positively skewed



(mode < median < mean) distribution, containing a potential second peak within $[-18, -12]$ dBZ. $f_{Z_e}^{\mathrm{CD}}$ for PA shows two peaks at $-22$ dBZ and $-5$ dBZ, where the latter one is likely associated with ice or drizzle that are misclassified as CD. $f_{Z_e}^{\mathrm{noCD}}$ show positive skewed distributions for both sites, centered at $-18$ dBZ for PA and $-20$ dBZ for LE.

(B) radar mean Doppler velocity $\bar{v}_D$ in units of $[\mathrm{m\,s^{-1}}]$:

   $f_{v_D}$ shows similar morphologies (e.g. center values in $[-1, 0]$, mono-modal) for all classes and both sites. Due to the
orographically induces gravity waves at PA, the distributions show larger variance, where the peak in $f_{v_D}^{\mathrm{TP}}$ is located closer to $0\,\mathrm{m\,s^{-1}}$.

(C) $\log_{10}$ of lidar attenuated backscatter coefficient $\beta_{\mathrm{att}}$ in units of $[\log_{10}(\mathrm{m^{-1}\,sr^{-1}})]$: $f_{\beta_{\mathrm{att}}}^{\mathrm{TP}}$ and $f_{\beta_{\mathrm{att}}}^{\mathrm{FN}}$ distributions match very well for both LE and PA, showing negative skewness with peaks between $-4.7$ and $-4.4$, respectively. Most $f_{\beta_{\mathrm{att}}}^{\mathrm{TN}}$ correspond to low $\beta_{\mathrm{att}}$ values (expected for ice crystals), were PA data shows a peak at $-7$, which is noticeably lower
than the $f_{\beta_{\mathrm{att}}}^{\mathrm{TN}}$ peak for LE data at $-5.2$. $f_{\beta_{\mathrm{att}}}^{\mathrm{FP}}$ show distributions with peaks between $[-5.0, -5.5]$ where potentially larger amounts of small ice crystals were observed. Note that $f_{\beta_{\mathrm{att}}}^{\mathrm{CD}}$ and $f_{\beta_{\mathrm{att}}}^{\mathrm{noCD}}$ refer to lidar observations beyond thin liquid layers. Those data points are influenced by attenuation effects from liquid layers in lower altitudes and thus were excluded from analysis.

(D) $\log_{10}$ of eddy dissipation rate $\varepsilon_{\mathrm{DR}}$ in units of $[\mathrm{m^2\,s^{-3}}]$:

$f_{\varepsilon_{\mathrm{DR}}}^{\mathrm{TP}}$ is centered at $-3.5$ and $f_{\varepsilon_{\mathrm{DR}}}^{\mathrm{TN}}$ at $-4.5$ for both PA and LE. However, the peak in the noCD distribution shows the maxima at approx. $-5.5$. Assuming that liquid bearing layers are usually more turbulent than precipitating ice crystals (Luke et al., 2010), a peak at lower values in $f_{\varepsilon_{\mathrm{DR}}}^{\mathrm{noCD}}$ predictions is expected. $f_{\varepsilon_{\mathrm{DR}}}^{\mathrm{FP}}$ shows its peak at large values of $-3.8$, matching the peak of $f_{\varepsilon_{\mathrm{DR}}}^{\mathrm{FP}}$. $f_{\varepsilon_{\mathrm{DR}}}^{\mathrm{CD}}$ and $f_{\varepsilon_{\mathrm{DR}}}^{\mathrm{FN}}$ present a good fit. Overall, LE shows higher turbulence values than PA.

(E) radar linear depolarization ratio LDR in units of $[\mathrm{dB}]$: The LDR distribution for PA $f_{\mathrm{LDR}}$ shows two distinct peaks at
$-30$ (spherical particles) and $-20$ (columns) and three peaks at $-30$ (spherical), $-20$ to $-15$ (columns), and $-5$ (insects embedded in clouds) for LE.

(F) model temperature $T$ in units of $[\,^{\circ}\mathrm{C}]$:

   The distributions of values in $\mathcal{C}$ (Eq. 7) and CD show single peaks centered between $-10$ and $0$. $f_T^{\mathrm{noCD}}$ spreads over a larger $T$ range of $[-60, 20]$ that are indicative of regions where ice particles (noCD) are detected for $T < 0$ and drizzle
or rain at $T > 0$.

*Author contributions.* The method was developed by WS with contributions by HKL and TV. PS provided the DACAPO-PESO ceilometer and MWR data. WS performed data visualization and analysis with help from TV, HKL, MM, and PS. The text was written by WS and reviewed by all co-authors.





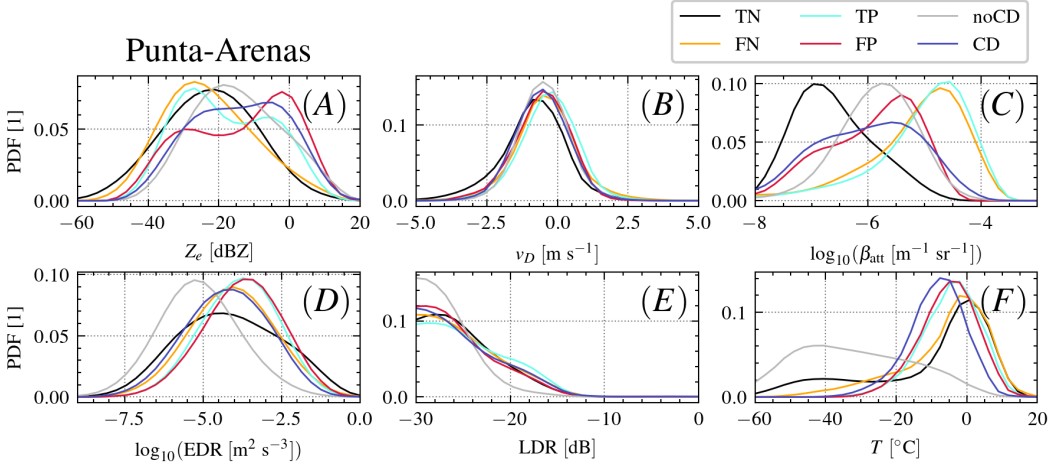

**Figure B1.** Probability density functions (PDFs) for different variables obtained in Punta Arenas, Chile.

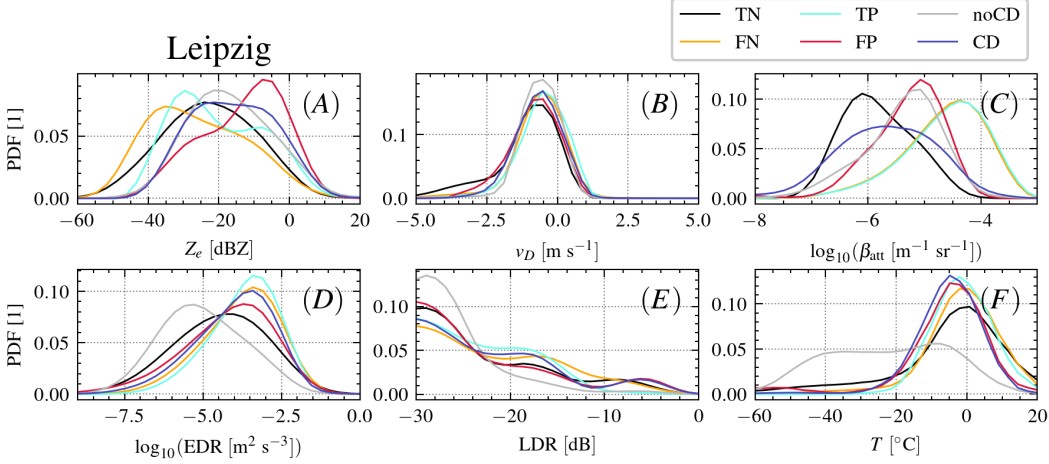

**Figure B2.** Probability density functions (PDFs) for different variables obtained in Leipzig, Germany.



*Competing interests.* MM and PS are associate editor of AMT. The peer-review process was guided by an independent editor, and the authors
have no other competing interests to declare.

*Code and data availability.* The raw data and Cloudnet data sets are provided by the ACTRIS Data Centre node for cloud profiling via the
following link (https://cloudnet.fmi.fi/). The machine learning results will be incooperated into the Cloudnet data processing chain and will
also be available via ACTRIS in the future. Meanwhile, they can be reprocessed via VOODOO (https://doi.org/10.5281/zenodo.5970206)
or obtained upon request from willi.schimmel@uni-leipzig.de. A setup of pyLARDA (https://doi.org/10.5281/zenodo.4721311, Bühl et al.
(2021)) was used for data input and analysis. Cloudnet processing was done using CloudnetPy (https://doi.org/10.5281/zenodo.4011843,
Tukiainen et al. (2020)).

*Acknowledgements.* This measure is co-financed by tax funds on the basis of the budget passed by the members of the Saxon state parliament.
We acknowledge the provision of physical access to the LACROS resources in the frame of DACAPO-PESO, which is provided via the
European Research Infrastructure for the observation of Aerosol, Clouds and Trace Gases (ACTRIS; grant nos. 654109 and 739530) as
part of the European Union's Horizon 2020 research and innovation programme. We also acknowledge ECMWF for providing Integrated
Forecasting System(IFS) model data. Further, we want to thank Boris Barja from the Universidad de Magallanes, for granting access to the
site and his support throughout the field campaign. We acknowledge ACTRIS for providing the Cloudnet framework used in this study, which
was developed by the Finnish Meteorological Institute (FMI), and is available for download from https://cloudnet.fmi.fi/. Special thanks to
Simo Tukiainen (FMI) for Cloudnet support and Teresa Vogl and Martin Radenz (TROPOS) for their remote-sensing and programming
knowledge. Also, thanks Marika Kaden (Mittweida University of Applied Sciences) for the advice in machine learning.

*Financial support.* This research has been supported by the Federal State of Saxony and the European Social Fund (ESF) in the framework
of the programme "Projects in the fields of higher education and research" (grant no. 100339509) and ESF-REACT (grant no. 100602743).
Parts of the DACAPO-PESO campaign were funded by the Deutsche Forschungsgemeinschaft (DFG – German Research Foundation) project
PICNICC (SE2464/1-1 and KA4162/2-1).



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
