# Peer review of "Identifying cloud droplets beyond lidar attenuation from vertically-pointing cloud radar observations using artificial neural networks"

_Atmospheric Measurement Techniques, 2022_

## Referee Comment (RC1)

**Major comments:**

Figure1: Does this mean the CNN algorithm can only detect cloud droplets in the presence of bi-modal spectra? I noticed the identified CD aera in Fig.7, Figure A4 are associated with this bimodality. If so, this may be related to the relative low "recall" number shown in Fig 9 and Fig 11 as the algorithm may tend to miss the scenario when cloud droplets exist but manifested as single peak spectrum (e.g., when the ice-contributed Doppler spectra is not distinguishable with the cloud Doppler spectra).

Line 243: I'm thinking the microphysics and the related process may be different in these two sites. Do you expect the proposed machine learning algorithm would generate a better performance if it was trained for these two sites separately?

Line 172-Line 175: Besides the "vertical generalization", have you considered the normalization horizontally? Specifically, I am wondering the effect of vertical air motion on the model performance. If the trained model is very sensitive to the spectra peak location, then the spectra location offset caused by the air motion may confuse the ML model.

Line 413-Line 415: I'm thinking some other quantities, like root mean square error, are more preferable to indicate the LWP difference between observation and retrieval. Please elaborate more on the reasoning of using correlation coefficient.

**Minor Comments:**

Line 243: I would tend to think more datasets are preferable for model training if the training datasets are properly cleaned. Did you notice the prediction performance reduced when more training dataset are included?

Line 40: remove e.g.

Line 106-Line 107: The conclusion "... is used to distinguish between cloud droplets and aerosol in Cloudnet" is not supported by the reasoning discussed above.

Line 150: The frequency modulation unit here (MHz) is not consistent with the one shown in Table3 (kHz), please confirm.

Line 157: Do you mean "polarized signals"?

Line 162: Please check this sentence.

Line 189: I couldn't find the precipitation rate products in the datasets, please add this information.

Line 202: Do you mean the "... if the detected cloud top by **lidar** was less than 500m above…"?

Line 214: This sentence should read like "Fig. 3 shows the architecture of the VOODOO retrieval algorithm"

For Figure 3: For the input, if I understand correctly, six Doppler spectra are constructed as a spectrogram for training. However, line 177 indicates that 30 time steps are used. Please confirm the number is consistent.

Line 361: Figure6 has no subplot.

Line 399: Please check this sentence: "… range are thatight than 0.8"

Line 400: Are you referring to the third column of Fig. 9?

Figure11: Please also add the probability density functions of LWP of the same datasets as shown in FigureB1 and B2, so the readers can see the relative frequency of the performance scores.

Line 426: Recall is not directly estimated by TN, please check the explanation here is reasonable.

---

## Author Response (AR1)

**VOODOO paper review**

Link: https://amt.copernicus.org/preprints/amt-2022-149/

**Discussion open (until 28 Jul 2022)**

Reviewer 1:

I want to thank the authors for this well-conceived and ably executed study. The proposed algorithm is well designed and is important for the mix-phased cloud studies. The manuscript is clearly structured, although there are some minor typo/grammar issues. With minor changes to address my comment below, I recommend this paper for publication in AMT. Please find the comments in the attachment,
Best,
Please also note the supplement to this comment:
https://amt.copernicus.org/preprints/amt-2022-149/amt-2022-149-RC1-supplement.pdf

**Major comments:**
Figure1: Does this mean the CNN algorithm can only detect cloud droplets in the presence of bi-modal spectra? I noticed the identified CD aera in Fig.7, Figure A4 are associated with this bimodality. If so, this may be related to the relative low "recall" number shown in Fig 9 and Fig 11 as the algorithm may tend to miss the scenario when cloud droplets exist but manifested as single peak spectrum (e.g., when the ice-contributed Doppler spectra is not distinguishable with the cloud Doppler spectra).
Although we cannot say with certainty which morphological features are used by the CNN to identify cloud droplets, we can see that the calculated probability for the presence of cloud droplets in bi-modal spectra is very high (>0.8 see Fig. 7 right). In Fig. A4 on the right, it can be seen that the CNN shows increased cloud droplet probability values at the cloud top (>0.3) even though the radar Doppler spectrum is monomodal. Here the CNN seems to be oriented to other features (possible spectrum width, skewness, or similar). The same is true for Fig. 7 left above 2.8 km and Fig 7 right, between 3-3.3 km.

Line 243: I'm thinking the microphysics and the related process may be different in these two sites. Do you expect the proposed machine learning algorithm would generate a better performance if it was trained for these two sites separately?
In short, yes. In the future, it will be possible for users to train the CNN model separately for the respective operational areas. Site specific training set will most likely enhance the performance. However, since the performance scores for the second site (Leipzig) are also high, we concluded that no retraining of the CNN for this dataset was required. Kalesse-Los et al., 2021 (https://doi.org/10.5194/amt-15-279-2022) had tested the applicability of a CNN pre-trained in the Arctic on mid-latitudinal clouds and also found it to perform sufficiently well.

Line 172-Line 175: Besides the "vertical generalization", have you considered the normalization horizontally? Specifically, I am wondering the effect of vertical air motion on the model performance. If the trained model is very sensitive to the spectra peak location, then the spectra location offset caused by the air motion may confuse the ML model.
The Punta Arenas data set is very strongly influenced by orographic gravity waves (see Radenz et al., 2021; https://doi.org/10.5194/acp-21-17969-2021). Longer updrafts and down-drafts cause the location of the peaks to change significantly. Nevertheless, we were able to obtain good results in the prediction of cloud droplets for Punta Arenas. I.e. the CNN seems to generalize to the extent that this phenomenon does not have a major impact on the prediction.

Line 413-Line 415: I'm thinking some other quantities, like root mean square error, are more preferable to indicate the LWP difference between observation and retrieval. Please elaborate more on the reasoning of using correlation coefficient.
Added this information on Line 420-425: " Note that the method of \cite{Karstens1994} uses the adiabatic assumption to calculate the liquid water path in combination with the liquid water mask (time-height-mask for droplets presence), and ECMWF weather model data (temperature, air pressure, and specific humidity). Since the adiabatic assumption is not suitable in all cloud situations and the model data is subject to uncertainties, the idea was to compare only the correlation of both time series (MWR-LWP vs. LWP$_\mathrm{ad}$ of Cloudnet vs. LWP$_\mathrm{ad}$ of VOODOO). Therefore, we decided not to compare absolute values. "

**Minor Comments:**

Line 243: I would tend to think more datasets are preferable for model training if the training datasets are properly cleaned. Did you notice the prediction performance reduced when more training dataset are included?
We've noticed a drop in the recall score, when using more (e.g. 80% training vs. 20% validation) data for training. The distribution is 5 to 1 for noCD to CD samples. Arguably we assume, if the CNN is trained on more data, the CNN focuses also more on the noCD class.

Line 40: remove e.g.  removed

Line 106-Line 107: The conclusion "... is used to distinguish between cloud droplets and aerosol in Cloudnet" is not supported by the reasoning discussed above.   reformulate: "Due to its very high sensitivity to cloud droplets, beta_att is used in the Cloudnet processing to identify cloud droplets."

Line 150: The frequency modulation unit here (MHz) is not consistent with the one shown in Table3 (kHz). changed unit to kHz Line 150

Line 157: Do you mean "polarized signals"? yes, fixed typo

Line 162: Please check this sentence.
fixed sentence: "The radar moments $Z_e$, $\bar{v}_D$, $\sigma_w$, and LDR are estimated from the spectra and stored as NetCDF files. Those files are used as input for the Cloudnet processing."

Line 189: I couldn't find the precipitation rate products in the datasets, please add this information. precip rate from Vaisala WXT536 Compact Weather Station mounted to the cloud radar, I added this information to the manuscript Line 189

Line 202: Do you mean the "... if the detected cloud top by **lidar** was less than 500m above…"?
No, by default Cloudnet extends a liquid layer to cloud top using the cloud top as it was observed by radar. If this radar cloud top is less than 500m above the liquid cloud base (detected by lidar), Cloudnet classifies those pixels between liquid cloud base and radar cloud top as liquid, even though no lidar signal is available.

Line 214: This sentence should read like "Fig. 3 shows the architecture of the VOODOO retrieval algorithm" changed the sentence accordingly

For Figure 3: For the input, if I understand correctly, six Doppler spectra are constructed as a spectrogram for training. However, line 177 indicates that 30 time steps are used. Please confirm the number is consistent.
Clarified that for the radar settings used, six spectra are concatenated per sample. Line 176

Line 361: Figure6 has no subplot. removed (A) and (B) in text.

Line 399: Please check this sentence: "… range are thatight than 0.8"
rephrase sentence: "The first column of Fig. \ref{fig:s-metrix} shows that the precision values have a clustering close to 1"

Line 400: Are you referring to the third column of Fig. 9? yes, change to "third"

Figure11: Please also add the probability density functions of LWP of the same datasets as shown in FigureB1 and B2, so the readers can see the relative frequency of the performance scores.
This information is implicitly presented in Figure 9. B1 and B2 are PDFs  of 2D variables, however the LWP is time-series data. We could add the LWC (liquid water content), which is a 2D variable, although this is likely not very meaningful because it relies on many assumptions made in the Frisch et al.1994 retrieval; mainly the adiabatic assumption, but also model temperature, pressure, humidity uncertainty, etc.)

Line 426: Recall is not directly estimated by TN, please check the explanation here is reasonable. correct, changed TN to FP

**Reviewer 2:**

This paper presents a deep convolutional neural network (CNN)-based retrieval method (i.e., VOODOO) to analyze radar Doppler spectra to identify the probability of the existence of supercooled liquid in vertical radar columns. The training of the CNN was realized using the Cloudnet processing suite. Both case studies and long-term predictions of 18 months in total of cloud observations at two mid-latitude locations (Punta Arenas and Leipzig) are used to test and evaluate the retrieval method. Results show that VOODOO achieves good precision and accuracy and is best for multi-layer stratiform and deep mixed-phase cloud situations. VOODOO also shows a better correlation between MWR-based LLT and LWP, compared with Cloudnet which is limited in deep mixed-phase clouds due to lidar attenuation. The authors clearly present their methodology and the results are convincing. I have one general comment and some minor comments below.

**General comment:**

Bi-modal spectra in Figure 1 nicely separate the fast-moving ice particles and slow-moving cloud droplets. But it is not clear to me how VOODOO identifies the presence of cloud droplets if the radar Doppler spectra are single-mode but skewed or if the spectra have multiple modes (more than 2). Slowly falling secondary ice particles might also generate a peak in the Doppler spectra (e.g., Luke et al., PNAS, 2021).
Around Line 370, it said that "Below 2.6 km, smaller ice crystals are falling out of the mixed-phase cloud top, which are melting and form drizzle drops at approximately 1 km altitude." However, the temperature at about 1 km at that time is still below 0 C (Fig 5). Is it possible the bimodal spectra are due to the existence of fast-moving ice particles and slow-moving secondary ice particles instead of drizzle?

Luke, Edward P., Fan Yang, Pavlos Kollias, Andrew M. Vogelmann, and Maximilian Maahn. "New insights into ice multiplication using remote-sensing observations of slightly supercooled mixed-phase clouds in the Arctic." Proceedings of the National Academy of Sciences 118, no. 13 (2021): e2021387118.
change sentence (starting @ Line 370): "Below 2.6 km, smaller ice crystals are falling out of the mixed-phase cloud top. Multiple ice populations are present at altitudes between 0.7 – 1.8 km, indicated by the skewed Doppler spectra (see Fig. 7A) which separates into a bi-modal distribution between 1.1 – 0.7 km. This bimodality might be caused by secondary ice production at temperatures below -10°C \cite{Luke2021}. At 0.7 km the ice crystals melt and form drizzle drops with Doppler velocities below 2 ms-1 (see Fig. 5 B)."

Also see multichannel Raman polarization lidar PollyXT plots in Fig. X below: Colocated PollyXT raman lidar during 1. Aug. 2019 in Punta Arenas with focus on 2 – 4 UTC. Aerosol layer visible in att_bsc 532 and 1064 channel below 1 km altitude. PollyXT captures the few larger precipitating ice crystals below the liquid bearing layer at ~3 km altitude (light blue streaks), which are also visible in the linear volume depolarization (532 channel) with low values of ~6% depolarization. Ergo, low amounts of larger non-spherical particles can be assumed. However, below 1 km altitude the particles become spherical (depol < 2 %).

**Minor comment:**

1. Line 343: Remove bracket around Fig. 4D       removed brackets

2. Line 400: "second columns"->"third columns"?   yes, changed to "third"

3. Line 444: add "LLT (XXX)" after "MWR-based"?
 rephrase sentence, see line 443-445: "Nevertheless, for the long-term observations, VOODOO achieves good precision $(>0.60)$ and accuracy $(>0.73)$, confirmed indirectly by a strong correlation between MWR-LWP and LWP$_\mathrm{ad}$ $(>0.45)$ compared to the correlation with LWP$_\mathrm{ad}$ of Cloudnet $(<0.22)$. "

4. Line 445: "LWP (>0.63)". 0.63 is not consistent with the value before. Please check. changed to 0.45

Figure X